psychology

weight, body dissatisfaction, eating disorders, perception, body size, adaptation

# Does repeatedly viewing overweight versus underweight images change perception of and satisfaction with own body size?

Helen Bould[1,2,6], Katharine Noonan[2],
Ian Penton-Voak[4,5,8], Andy Skinner[4],
Marcus R. Munafò[3,4,5], Rebecca J. Park[2,6],
Matthew R. Broome[2,5,6] and Catherine J. Harmer[2,7]

[1]Centre for Academic Mental Health, Population Health Sciences, University of Bristol, Oakfield House, Oakfield Grove, Bristol BS8 2BN, UK
[2]Department of Psychiatry, University of Oxford, Warneford Hospital, Warneford Lane, Oxford OX3 7JX, UK
[3]UK Centre for Tobacco and Alcohol Studies, School of Psychological Science, University of Bristol, 12a Priory Road, Clifton, Bristol BS8 1TU, UK
[4]MRC Integrative Epidemiology Unit, University of Bristol, Oakfield House, Oakfield Grove, Bristol BS8 2BN, UK
[5]National Institute of Health Research Biomedical Research Centre at the University Hospitals Bristol NHS Foundation Trust and the University of Bristol, Bristol, UK
[6]Institute for Mental Health, School of Psychology, College of Life and Environmental Sciences, University of Birmingham, Birmingham, UK
[7]Oxford Health NHS Foundation Trust, Warneford Hospital, Oxford, UK
[8]School of Psychological Science, University of Bristol, 12a Priory Road, Clifton, Bristol BS8 1TU, UK

 HB, 0000-0001-8163-3210; IP-V, 0000-0003-4232-0953;
MRM, 0000-0002-4049-993X; MRB, 0000-0002-6963-8884;
CJH, 0000-0002-1609-8335

**Author for correspondence:**
Helen Bould
e-mail: helen.bould@bris.ac.uk

Body dissatisfaction is associated with subsequent eating disorders and weight gain. One-off exposure to bodies of different sizes changes perception of others' bodies, and perception of and satisfaction with own body size. The effect of repeated exposure to bodies of different sizes has not been assessed. We randomized women into three groups, and they spent 5 min twice a day for a week completing a one-back task using images of women modified to appear either under, over, or neither over- nor underweight. We tested the effects on their perception of their own and others' body size, and satisfaction with own size. Measures at follow-up were compared between groups, adjusted for baseline measurements. In 93 women aged

18–30 years, images of other women were perceived as larger following exposure to underweight women (and vice versa) ($p < 0.001$). There was no evidence for a difference in our primary outcome measure (visual analogue scale own size) or in satisfaction with own size. Avatar-constructed ideal ($p = 0.03$) and avatar-constructed perceived own body size ($p = 0.007$) both decreased following exposure to underweight women, possibly due to adaptation affecting how the avatar was perceived. Repeated exposure to different sized bodies changes perception of the size of others' bodies, but we did not find evidence that it changes perceived own size.

## 1. Introduction

Body dissatisfaction is an important risk factor for both subsequent eating disorders [1,2] and weight gain [3]. New strategies to prevent both eating disorders and weight gain are urgently needed, since eating disorders are associated with very high mortality [4,5] and obesity rates are increasing globally [6]. More than one in three adults in the USA [7] and 29% of adults in England [8] now meet criteria for obesity. Body dissatisfaction is a potentially modifiable target for both prevention and treatment.

Body dissatisfaction is often postulated to have two components—perception of own size, and an attitudinal or cognitive component of dissatisfaction with body size and shape [9]. This gives rise to the idea that if it were possible to change perception of own size, it might be possible to change satisfaction with size. There is growing body of literature suggesting that it is possible to change participant perception of *others*' body size, simply by exposing them to images of bodies of different sizes. The body size viewed as being most 'normal' becomes thinner after exposure to thinner bodies, and larger following exposure to larger bodies [10–17]. Viewing photographs of males whose body mass index (BMI) is in the obese, as opposed to normal, category leads to both male and female participants judging an image of a man with a BMI in the overweight range to have a healthy, normal weight [13,14]. The body size perceived as most attractive also becomes smaller following exposure to thinner as opposed to larger bodies [10,11].

The literature around whether such interventions might also change perception of own size is much more limited. Three studies ($N = 14$, 16 and 29) found exposure to thinner-than-actual photographs of self or others led to participants subsequently judging a thinner-than-actual photograph of self as more accurate; those shown fatter-than-actual photos subsequently thought fatter-than-actual photos of self were more accurate [16–18]. Brooks and colleagues [12] found that exposure to images of either thinner versions of self or thinner versions of others led to subsequent images of self and others being viewed as larger (and vice versa). However, although the effects did transfer to some extent between images of own and others' bodies, the effects were larger when the exposure and testing were with the same body type (e.g. both images were of self). Previous work from our group [19] found that exposure to images of women stretched to appear underweight, overweight or normal weight altered participants' view of their own actual size, as viewed in a mirror, and their satisfaction with own size: those exposed to overweight images subsequently viewed themselves as smaller and were more satisfied with their size, compared to those who saw the same women manipulated to appear underweight.

Limitations of these previous studies include small sample size [16–18], and images lacking in ecological validity since they were uniformly modified to appear of different sizes rather than manipulated in line with typical patterns of weight gain [19]. Further possible limitations of previous studies are their reliance on measures that are susceptible to bias due to demand characteristics: that is, participants inferring the aim of a study and trying to be 'good subjects' by attempting to respond in the way they believe will assist the researchers [20]. The key difference between previous studies and the study reported here, is that previous studies have involved a single session of exposure to the images, whereas our study involves repeated sessions of exposure over the course of a week. One would anticipate that any proposed intervention to address body dissatisfaction would need to be given repeatedly, in order to counteract ongoing exposure to underweight images in visual media; relatedly, in order to understand the effects of exposure to media images, we need to study the effects of repeated, as well as one-off, exposure to images of others' bodies.

The current study was designed to test the effect of repeated exposure (5 min twice a day) over the course of a week to images of women of different sizes on women's perception of their own and others' size, and satisfaction with own size. We used images from the Morphed Photographic Figure Scale [21], manipulated using morph techniques to provide realistic representations of changes in women's body shape with changes in weight (figure 1).

Participants were randomized into three groups, to train using images of the same women modified to appear either 'underweight', 'overweight' or 'neither over- nor underweight'. Our hypotheses, based on the literature to date, were that repeated training using 'overweight' as opposed to 'normal' or

**3**

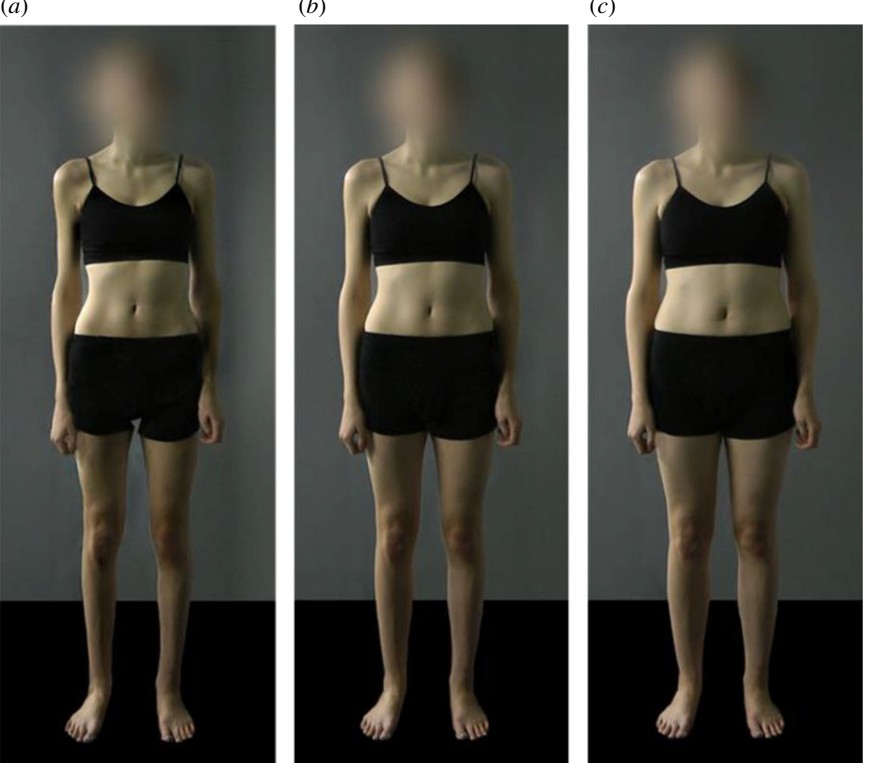

**Figure 1.** Examples of stimuli used: (*a*) 'underweight'; (*b*) 'neither over- nor underweight'; (*c*) 'overweight'.

'underweight' images would lead to participants (i) viewing images of others' bodies as smaller, and vice versa; (ii) viewing their own bodies as smaller, and vice versa; (iii) feeling more satisfied with their bodies; (iv) having a larger ideal size, and (v) that results would be consistent across a range of explicit, behavioural and implicit measures. We pre-registered our primary outcome measure of perceived own size. We also included additional outcome measures, including measures designed to enable participants to show us the perceived size and shape of their current and ideal body using an avatar, to capture behavioural changes related to body satisfaction, and exploratory measures to assess implicit perception of and satisfaction with own body size.

## 2. Methods

### 2.1. Participants

We recruited participants by e-mail advertisement (to undergraduate and graduate students at the University of Oxford via college junior and middle/graduate common rooms and University departments), word of mouth, posters in departments and colleges, and advertisements on Facebook. Potentially interested participants were asked to complete a brief online screening questionnaire to ensure that they met inclusion criteria.

Inclusion criteria required participants to be female, since our images were of female bodies, aged 18–30 years old, speak English as a first language (since some of the tasks required this) and have a BMI within the healthy range of 18–25 kg m$^{-2}$, calculated from their self-reported weight and height. Participants were excluded if they had a current or previous eating disorder, were undergoing treatment for any other mental illness or taking psychoactive medication that may affect cognitive function (e.g. street/illicit drugs, methylphenidate, St John's Wort), had any current serious neurological illness (e.g. MS, epilepsy), had coeliac disease, were currently pregnant, smoked more than 10 cigarettes per day or had dyslexia. Participants were offered £60 to thank them for their time.

The study was approved by the Oxford University Central University Research Ethics Committee (MS-IDREC-C1-2015-083). The study protocol was pre-registered on the Open Science Framework (https://osf.io/pwd36/).

## 2.2. Measures

At both the baseline session (before training) and the follow-up session (after training), participants completed the following range of measures of their perception of and satisfaction with their body size.

### 2.2.1. Explicit measures

The pre-specified primary outcome measure was scored on a 10-point visual analogue scale (VAS) measure of own size from too thin (0) to too fat (10) (in response to the question 'Please indicate on this scale what you feel is the size of your body at the moment'), and the secondary outcome measure was VAS satisfaction with own size from very dissatisfied (0) to very dissatisfied (10) (in response to the question 'Please indicate on this scale how satisfied you feel about the size of your body at the moment').

To measure perceived body size in others, participants rated a series of 90 images of bodies [21], similar to those used in the training task with regard to whether each body was over, under or neither over- nor underweight. Images were presented in a randomized order using EPrime software and participants used the relevant key on the keyboard to answer the question 'Is this woman over-weight (o), underweight (u) or neither over- nor underweight (n)?' Twenty-four of these images were used in the training task itself (eight in each randomization group), so each individual participant saw eight of these images during their training.

In order to obtain more detailed results with regard to perceived own size, and to investigate whether ideal size was altered by exposure to different-sized images, participants were also given 2 min to alter the weight, bust, waist and hip sizes of a computer avatar to make it their 'own size' (two attempts); and 'the size and shape you would most like to be' (ideal size) (two attempts). The avatar was provided by the company BodyLabs (https://shapex.bodylabs.com), and was created from a large database of real women's body shapes such that although they can be manipulated to different sizes, the body shapes created remained biologically plausible. Participants used four sliders, without numbers, to alter the avatar's size. In each version of the task ('own size' and 'ideal size'), they were presented with two sequential starting images, both of which had been created to be the same height as the participant: the first of these avatars began at their pre-reported body weight plus 10%, and the second at their body weight minus 10%. We used the weight and height measurements reported in the screening questionnaire. They were told that the avatar was their height but not their weight. They were shown the sliders and that the avatar could be rotated to view her from different angles, and they were asked to take up to two minutes to 'make this avatar your own size and shape', before clicking 'next' to have a second attempt. Once finished, they were asked to look away while the experimenter noted the size and shape they had made the women. The results were used to calculate 'Own size' avatar BMI and 'Ideal Own size' avatar BMI measures.

Participants also completed the following questionnaire measures: the body shape satisfaction scale (BSSS) [22] (a detailed measure of body satisfaction) and the eating disorder examination-questionnaire (EDE-Q) [23], (a marker of eating disorders with components specifically addressing weight concern and shape concern); the positive and negative affect scale (PANAS) [24], the state-trait anxiety inventory (STAI) (State) [25], the Beck depression inventory (BDI) [26] and the Rosenberg self-esteem scale (RSES) [27], to measure concurrent mood, anxiety and self-esteem, which may affect perception of own body, and may be altered as a result of changes in perception of own body. We also used a measure of handedness [28], in order to check that handedness was randomly distributed between groups, since some tasks required responding with left or right hand. Participants gave demographic information including their nationality, education level and media use. At the end of the follow-up session, participants were weighed and their height was measured.

### 2.2.2. Behavioural and implicit measures

We included additional exploratory measures in order to help address possible problems with demand characteristics of the study, thus reducing a potential source of bias, and to test whether any changes in reported beliefs led to changes in behaviour, or changes on implicit measures.

Participants were asked to rate a set of images of 60 outfits, ranging from swimwear to winter coats and trousers, as to 'How comfortable with your body would you feel wearing this outfit in an appropriate public setting?', from 1 (not at all comfortable) to 7 (very comfortable) (the outfit choice task) [29].

Halfway through the session, and after all reaction time tasks had been completed, participants were offered a cup of tea or coffee (or a glass of water) and a bowl containing 100 g of small, sweet, oat biscuits, and were given a 5 min break, during which the experimenter left the room. The quantity of biscuits consumed was measured after the participant had left, in order to measure food consumption in relation to body satisfaction.

Participants also completed four new implicit tasks for exploratory analysis of whether they were related to, or might potentially be more sensitive to change than, the explicit measures. These were (i) a lexical decision task (LDT) [30] comparing reaction times to words in the categories of 'beauty', 'ugly', 'positive' and 'neutral'; participants completed this task immediately after adjusting the avatar to their own shape and size, with the intent that this would mean that they were primed to be thinking about their own body during the LDT; (ii) a recall task of the numbers of words they could recall from the LDT; (iii) an implicit association test (IAT) involving sorting words relating to 'self' or 'other' and 'slim' or 'not slim'; and (iv) an IAT involving sorting words relating to self or other, and images of normal weight or slightly overweight women. These tasks are described at greater length in the electronic supplementary material.

## 2.3. Training task stimuli

The training task stimuli were chosen from the images in the morphed photographic figure scale [21]. We selected images for this study based on ratings of whether the images were underweight, overweight or neither over- nor underweight made by 170 of the subjects in the online study [21], who were drawn from an equivalent population to those recruited to the current study (i.e. female, BMI 18–25 kg m$^{-2}$, no current or previous eating disorder, aged 18–30 years). Eight images were chosen for each randomization group ('underweight', 'neither over- nor underweight', and 'overweight'), and the images in the three groups were of the same individual women, but modified to appear of different weights. The eight images for the 'neither over- nor underweight' group were chosen by selecting those versions of the images which most online participants rated as neither over- nor underweight. The eight images for the underweight group were chosen by selecting images that around half the online participants rated as underweight, and the eight for the overweight group from those that around half the online participants had rated as overweight. Images were chosen in this way as we did not want to engender strong emotional reactions to images at the more extreme ends of the scale, and we wanted the differences between groups to be subtle, such that subjects would be unaware of the purpose of the experiment. Example stimuli are shown in figure 1.

## 2.4. Procedure

Participants attended two experimental sessions each lasting between 1 and 1.5 h at the Department of Psychiatry, University of Oxford, on Day 1 and Day 8. Participants were asked to eat a light meal 2 h before attending, with the aim that satiety would be similar between the participants, and to avoid caffeine, alcohol and cigarettes in the 2 h before the session in order not to affect the reaction time tasks. At the start of Session 1, we took informed consent for the tasks they would be undertaking; participants remained blind to the hypotheses and purpose of the study. Participants were tested individually by one of two experimenters (H.B. and K.N.). They were randomized to one of the three groups using a computer-generated random sequence (group assignment determined whether each individual would be trained using images of overweight, underweight or neither over- nor underweight women). An individual not involved in any testing or participant contact assigned one of the three training conditions to each participant number, so both experimenters and participants were blind to training condition.

Participants completed the tasks described above, and were then briefly trained in how to use the laptop and the training task, before they took it home with them. They were asked to complete the 5 min training task on the laptop twice a day for a week. They were told that the laptop would record each trial, and they were invited to choose what times of the day would be convenient for them to complete the task. All participants agreed to receive automated text messages to their mobile phones at the times they had suggested, to remind them to complete the task.

At the end of the Day 8 session, participants were weighed and their height was measured.

## 2.5. The training task

The training task consisted of a one-back working memory task. The stimuli (photographs of eight women, altered as described above to appear either overweight, underweight or neither overweight nor underweight) were presented sequentially to participants, who were instructed to respond as to whether the image had the same identity as the one presented immediately before, or a different identity. This was to ensure that participants paid attention to the images displayed, which has been shown to enhance adaptation effects [31]. In the underweight condition, the images were all of women modified to appear underweight; in the overweight condition, the images were all of women modified to appear overweight; in the 'neither over- nor underweight' condition, the images were all of neither over- nor underweight women.

The training task was programmed using HTML, JavaScript and stylesheets. The program was designed to run in a web browser without an Internet connection, with the results stored offline within the browser. Each image was displayed for 2000 ms with a 20 ms inter-trial interval. To prevent low-level visual adaptation, the images were spatially jittered by an amount varying randomly from the centre of the screen by up to ±51 pixels in the x coordinate and ±38 pixels in the y coordinate (approx. 10% in the vertical and 10% in the horizontal plane of 1024 by 768 pixel display). Image size was 562 by 765 pixels, with a viewing distance of approximately 1 m. Failure to respond within the 2000 ms presentation resulted in no response being recorded for that trial. Each training block consisted of 48 trials in which eight images were the same as the preceding image. A total of three blocks, giving 144 individual trials, were presented, taking approximately 5 min to complete. Participants were given feedback on their accuracy halfway through and at the end of the session with the aim of encouraging them to try their best to complete the task. If participants did not make any responses during the session, they were prompted that they needed to respond.

## 2.6. Statistical analysis

We calculated means, standard deviations and median and interquartile ranges for both existing and novel measures at baseline, as well as descriptive data on age, educational background and social media use. These are presented at baseline for the entire sample, and by randomization group.

The primary outcome measure was VAS judgement of own body size at Session 2, adjusted for Session 1 score. The effects of training on judgement of own size were assessed using linear regression to compare post-intervention scores, adjusted for pre-intervention scores, by including the pre-intervention score as a covariate and using the Stata command 'regress'. Power calculation indicated that three groups of 30 would give us 90% power at an $\alpha$ level of 5% to detect an effect size of 0.38. All secondary outcome measures were assessed using regression analysis to compare Session 2 scores, each adjusted for relevant Session 1 scores. Randomization group was treated as a categorical measure throughout. Analyses were repeated after adjusting for potentially confounding variables: number of images seen during training (as a measure of the amount of exposure participants had to the different images—there was some variation since participants did not all comply with the instruction to complete the whole task, twice a day for 5 days), time since last training (in tertiles) (in case any effects reduce over time), measured BMI (which influences perception of body size of others [21]), and time spent using media (in tertiles) (which may negate or reduce the effect of the study images).

Analyses were completed using Stata 14. The data forming the basis of the results presented here are available on the University of Bristol Repository (https://data.bris.ac.uk/data/dataset/my46h06p89rw163w u2du2jjzn) [32].

# 3. Results

## 3.1. Descriptive data

We recruited 93 participants, who completed the study between 27 August 2015 and 22 April 2016. Study recruitment is shown in the CONSORT diagram (figure 2), and baseline descriptive data are presented in tables 1 and 2; electronic supplementary material, table S2. Data from the one-back task suggested that there were no between-group differences in terms of amount of time spent on the task, duration from last task completion to coming for follow-up appointment, or task accuracy. In terms of missing data, one person did not attend follow-up, and their results were not included in the analysis. We had missing data for one participant for perception of others' body size, and two participants for own and ideal body size, due to in-session problems with the programs for running these tasks. Analyses for these measures were conducted excluding those participants with missing data on these measures. We were unable to retrieve data from three laptops on the time elapsed since the last time the training was completed, and adjusted analyses were, therefore, run excluding these subjects.

## 3.2. Explicit measures

### 3.2.1. Primary outcome measures: VAS own size and satisfaction with own size

We found no evidence of a difference between training groups on our primary outcome measure of VAS own size (test of trend across groups: $p = 0.78$); nor was there any evidence of a difference between

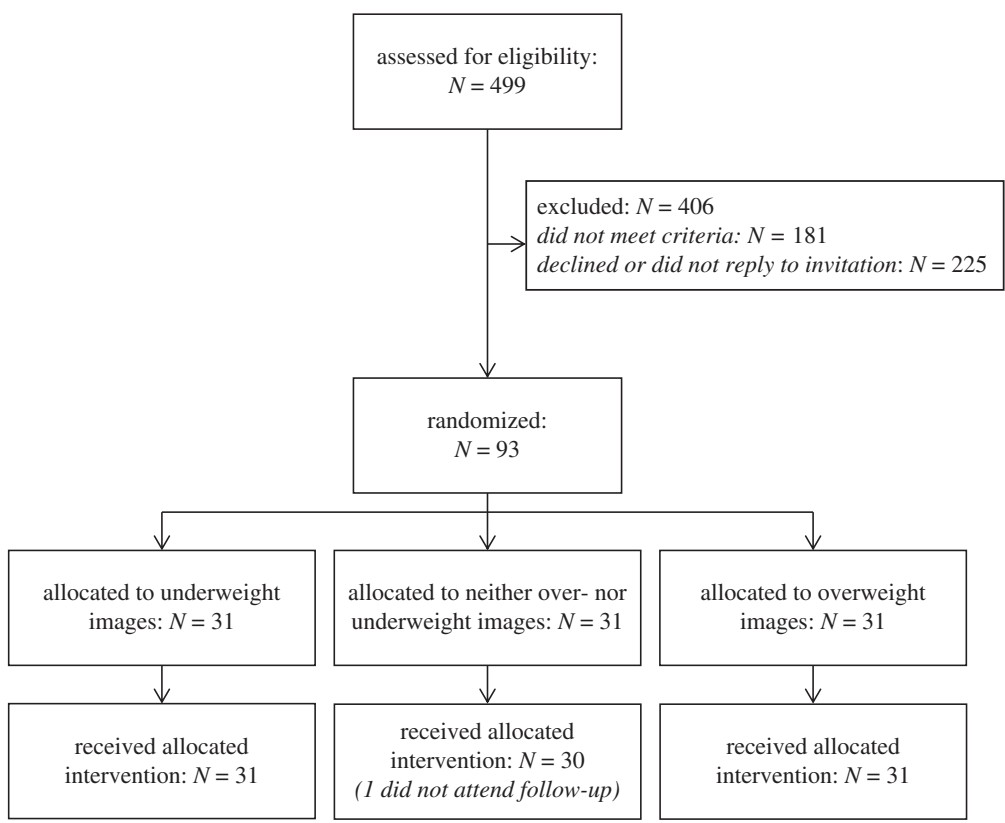

**Figure 2.** CONSORT diagram.

**Table 1.** Baseline descriptive data: comparison of groups.

| | all groups | adapted to underweight | adapted to neither over- nor underweight | adapted to overweight |
|---|---|---|---|---|
| age (mean, s.d.) | 22.7 (3.15) | 22.9 (3.57) | 23.4 (2.94) | 21.7 (2.73) |
| measured BMI (mean, s.d.) | 21.9 (2.11) | 21.9 (2.16) | 21.8 (2.20) | 22.1 (2.02) |
| degree-level education (N, %) | 45 (48.4) | 12 (38.7) | 18 (58.1) | 15 (48.4) |
| VAS satiety (0, very hungry; 10, very full) (mean, s.d.) | 5.9 (1.4) | 6.3 (1.4) | 5.8 (1.4) | 5.6 (1.4) |
| VAS own size from 0 (too thin) to 10 (too fat) (mean, s.d.) | 5.7 (0.94) | 5.5 (0.73) | 5.9 (1.07) | 5.7 (0.96) |
| VAS satisfaction with own size from 0 (very dissatisfied) to 10 (very satisfied) (mean, s.d.) | 6.1 (2.02) | 6.5 (1.8) | 5.8 (2.04) | 6.1 (2.2) |
| spend > 4 h a day using media (N, %) | 28 (30.1) | 10 (32.3) | 9 (29.0) | 9 (29.0) |
| number of Facebook friends (mean, s.d.) | 527 (295) | 525 (282) | 508 (305) | 547 (305) |
| right-handed N (%) | 77 (82.8) | 23 (74.2) | 26 (83.9) | 28 (90.3) |
| N images seen during training (mean, s.d.) | 1934 (259) | 1935 (198) | 1926 (273) | 1941 (302) |
| last training > 6 h before follow-up (N, %) | 30 (33.3) | 9 (30) | 11 (36.7) | 10 (33.3) |
| proportion correct responses in N-back (mean, s.d.) | 0.62 (0.02) | 0.62 (0.02) | 0.61 (0.02) | 0.62 (0.02) |

**Table 2.** Baseline descriptive data for additional questionnaire measures, avatar task and implicit measures.

| | all groups | adapted to underweight | adapted to neither over nor underweight | adapted to overweight |
|---|---|---|---|---|
| *N* computer images viewed as underweight (mean, s.d.) | 22.5 (8.38) | 23.6 (8.31) | 21.2 (9.26) | 22.7 (7.60) |
| *N* computer images viewed as overweight (mean, s.d.) | 30.2 (9.62) | 30.1 (8.52) | 32.1 (8.69) | 28.4 (11.3) |
| 'Own size' avatar BMI (mean, s.d.) | 22.1 (3.84) | 21.9 (4.17) | 21.8 (3.90) | 22.5 (3.53) |
| 'Ideal own size' avatar BMI (mean, s.d.) | 19.6 (3.00) | 20.0 (2.33) | 18.9 (2.89) | 19.9 (3.62) |
| BSSS (mean, s.d.) | 3.42 (0.70) | 3.51 (0.57) | 3.3 (0.64) | 3.44 (0.87) |
| EDE-Q (mean, s.d.) | 0.91 (0.88) | 0.63 (0.61) | 1.03 (0.99) | 1.08 (0.93) |
| PANAS (positive) (mean, s.d.) | 30.8 (6.68) | 30.7 (6.63) | 30.6 (6.29) | 31.1 (7.27) |
| PANAS (negative) (mean, s.d.) | 12.3 (2.82) | 12.1 (2.11) | 12.0 (2.61) | 12.7 (3.59) |
| STAI (mean, s.d.) | 28.9 (7.05) | 28.2 (7.39) | 28.2 (6.06) | 30.2 (7.63) |
| BDI (mean, s.d.) | 3.25 (3.86) | 3.00 (2.89) | 3.29 (3.91) | 3.45 (4.70) |
| RSES (mean, s.d.) | 22.4 (4.57) | 22.4 (3.95) | 22.3 (4.98) | 22.4 (4.86) |
| outfit choice task score (mean, s.d.) | 5.53 (0.88) | 5.37 (0.94) | 5.55 (0.94) | 5.68 (0.77) |
| IAT words (mean, s.d.) | 0.34 (0.43) | 0.44 (0.34) | 0.30 (0.45) | 0.29 (0.49) |
| IAT images (mean, s.d.) | 0.05 (0.28) | 0.03 (0.27) | 0.08 (0.32) | 0.02 (0.27) |
| LDT % recall of beauty words (mean, s.d.) | 47.2 (23.0) | 47.6 (27.1) | 41.4 (21.5) | 52.7 (19.0) |
| LDT % recall of ugly words (mean, s.d.) | 30.4 (19.4) | 27.1 (22.5) | 33.6 (16.9) | 30.5 (18.6) |

training groups in VAS satisfaction with own size (test of trend across groups: $p = 0.18$; table 3). This lack of between-group difference persisted following adjustment for the possible confounding variables described above.

### 3.2.2. Perception of others' body size

We found differences between training groups in the number of images of other women they perceived as being overweight or underweight ($p < 0.001$). Following training with underweight compared to neither over- nor underweight images, women judged 6.43 fewer images as being underweight (95% confidence interval [CI] −9.27, −3.59, $p < 0.001$), and 6.41 more images as overweight (95% CI 3.58, 9.24, $p < 0.001$). By contrast, following training with overweight images, women viewed 7.42 more images as being underweight (95% CI 4.62, 10.22, $p < 0.001$) and 5.92 fewer images as being overweight (95% CI −8.75, −3.09, $p < 0.001$) (see also figure 3a,b). These differences persisted following adjustment for potential confounding variables (number of images seen during training, time since last training (in tertiles), measured BMI, time spent using media (in tertiles); table 3).

### 3.2.3. Own and ideal size avatar BMI

There were differences between training groups in the BMIs of their 'own size' avatars (test of trend $p = 0.007$) and 'ideal size' avatars (test of trend $p = 0.03$) (figure 4a,b). Training with underweight images led to both a smaller ideal size, and also to a smaller perceived own size, in comparison with training with normal or overweight images. Differences in 'own size' persisted following adjustment for confounders; strength of evidence for differences in 'ideal size' was reduced (test of trend $p = 0.10$; table 3).

**Table 3.** Results for explicit tasks (see electronic supplementary material). Bold print indicates $p < 0.05$.

| | unadjusted comparison with those adapted to neither over- nor underweight images (Session 2 scores adjusted for Session 1 scores) (N = 92) | | | comparison with those adapted to neither over- nor underweight images, adjusted for number of images seen, time since last training, measured BMI, time spent using media (N = 89) | | |
|---|---|---|---|---|---|---|
| | adapted to underweight images | adapted to overweight images | test of trend across groups | adapted to underweight images | adapted to overweight images | test of trend across groups |
| VAS own size (N = 92) | 0.11 (−0.29, 0.51), $p = 0.58$ | −0.02 (−0.41, 0.38), $p = 0.94$ | $p = 0.78$ | 0.03 (−0.37, 0.44), $p = 0.87$ | −0.01 (−0.42, 0.39), $p = 0.96$ | $p = 0.97$ |
| VAS satisfaction with own size (N = 92) | 0.16 (−0.56, 0.88), $p = 0.66$ | −0.48 (−1.20, 0.24), $p = 0.19$ | $p = 0.18$ | 0.19 (−0.59, 0.98), $p = 0.63$ | −0.39 (−1.20, 0.42), $p = 0.34$ | $p = 0.33$ |
| N computer images viewed as underweight (N = 91) | **−6.43 (−9.27, −3.59), p < 0.001** | **7.42 (4.62, 10.2), p < 0.001** | **p < 0.001** | **−6.44 (−9.36, −3.51), p < 0.001** | **7.43 (4.41, 10.46), p < 0.001** | **p < 0.001** |
| N computer images viewed as overweight (N = 91) | **6.41 (3.58, 9.24), p < 0.001** | **−5.92 (−8.75, −3.09), p < 0.001** | **p < 0.001** | **6.51 (3.58, 9.44), p < 0.001** | **−5.88 (−8.94, −2.82), p < 0.001** | **p < 0.001** |
| 'Own size' avatar BMI (N = 90) | −0.17 (−1.21, 0.88), $p = 0.75$ | **1.40 (0.35, 2.44), p = 0.009** | **p = 0.007** | 0.05 (−1.04, 1.14), $p = 0.93$ (N = 88) | **1.64 (0.50, 2.78), p = 0.005** (N = 88) | **p = 0.006** |
| 'Ideal own size' avatar BMI (N = 90) | −0.31 (−1.03, 0.42), $p = 0.40$ | 0.64 (−0.09, 1.36), $p = 0.08$ | **p = 0.03** | −0.31 (−1.09, 0.46), $p = 0.43$ (N = 88) | 0.52 (−0.28, 1.34), $p = 0.20$ (N = 88) | $p = 0.10$ |

**Table 4.** Results for questionnaires (see electronic supplementary material). Bold print indicates p < 0.05.

| | unadjusted comparison with those adapted to neither over- nor underweight images (Session 2 scores adjusted for Session 1 scores) (N = 92) | | | comparison with those adapted to neither over- nor underweight images, adjusted for number of images seen, time since last training, measured BMI, time spent using media (N = 89) | | |
| | adapted to underweight images | adapted to overweight images | test of trend across groups | adapted to underweight images | adapted to overweight images | test of trend across groups |
| --- | --- | --- | --- | --- | --- | --- |
| BSSS | 0.04 (−0.13, 0.20), p = 0.65 | −0.07 (−0.23, 0.09), p = 0.40 | p = 0.41 | 0.04 (−0.13, 0.21), p = 0.64 | −0.06 (−0.24, 0.21), p = 0.52 | p = 0.52 |
| EDE-Q | 0.05 (−0.11, 0.20), p = 0.56 | 0.03 (−0.11, 0.18), p = 0.68 | p = 0.83 | 0.04 (−0.11, 0.19), p = 0.60 | 0.02 (−0.12, 0.18), p = 0.75 | p = 0.86 |
| STAI | 0.13 (−2.62, 2.88), p = 0.93 | 1.33 (−1.44, 4.10), p = 0.34 | p = 0.58 | −0.08 (−2.95, 2.79), p = 0.96 | 0.56 (−2.44, 3.56), p = 0.71 | p = 0.89 |
| PANAS positive | 1.57 (−0.97, 4.11), p = 0.22 | −1.50 (−4.04, 1.04), p = 0.24 | p = 0.06 | **1.22 (−1.29, 3.73), p = 0.34** | **−1.85 (−4.47, 0.76), p = 0.16** | **p = 0.05** |
| PANAS negative | −0.40 (−2.15, 1.35), p = 0.65 | 0.83 (−0.92, 2.59), p = 0.35 | p = 0.36 | −0.44 (−2.23, 1.33), p = 0.62 | 0.46 (−1.39, 2.32), p = 0.62 | 0.59 |
| RSES | **0.53 (−0.81, 1.87), p = 0.44** | **−1.14 (−2.48, 0.20), p = 0.10** | **p = 0.04** | 0.46 (−0.90, 1.83), p = 0.50 | −0.99 (−2.40, 0.43), p = 0.17 | p = 0.11 |
| BDI | 0.19 (−1.21, 1.60), p = 0.79 | 0.16 (−1.25, 1.56), p = 0.82 | p = 0.96 | 0.09 (−1.19, 1.38), p = 0.89 | −0.72 (−2.06, 0.62), p-0.29 | p = 0.41 |

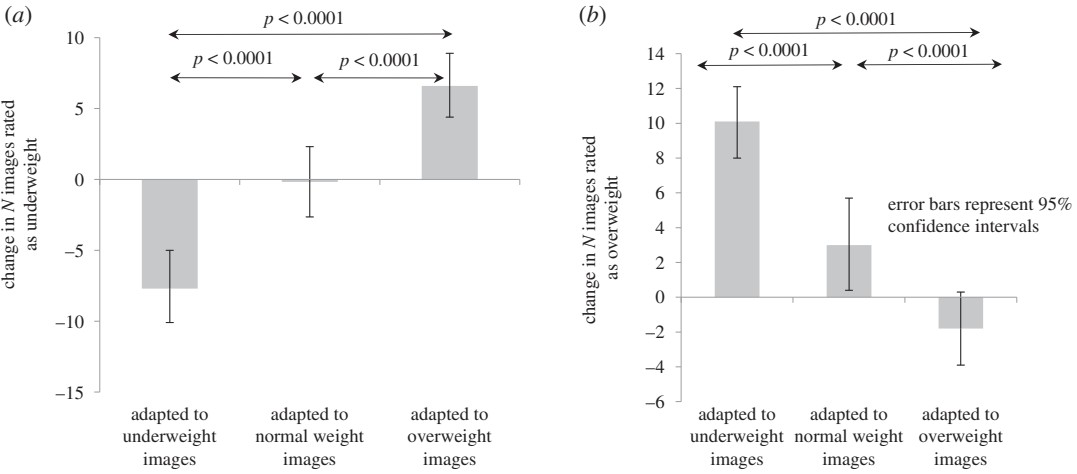

**Figure 3.** Change in number of others' bodies rated as 'underweight' or 'overweight'.

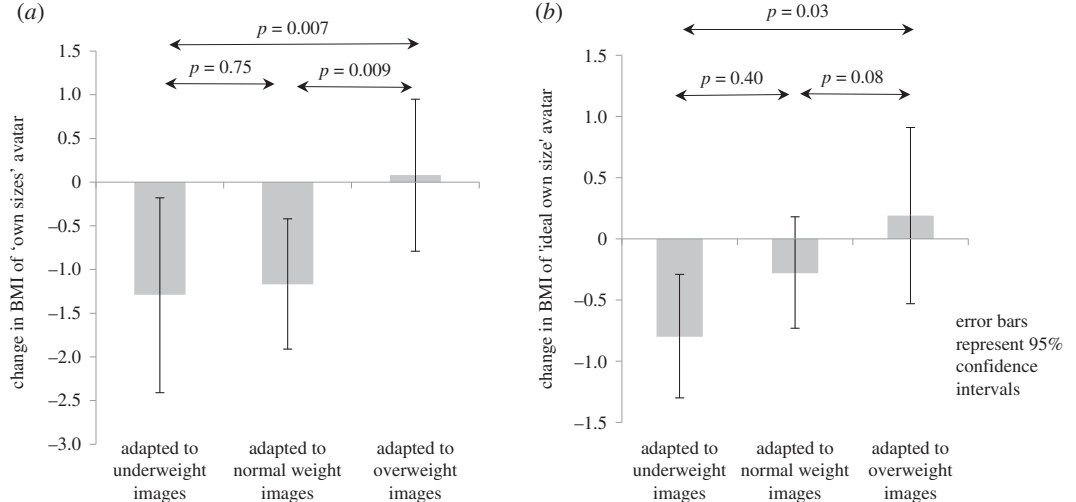

**Figure 4.** Change in own and ideal size, as represented with the avatar.

### 3.2.4. Questionnaire measures

The only between-group difference seen in the questionnaires was weak evidence for higher PANAS positive scores following training with smaller images (test for trend $p = 0.06$); this was very slightly strengthened following adjustment for possible confounders (test for trend $p = 0.05$). There was also a suggestion of higher self-esteem scores following training with underweight images (test for trend $p = 0.04$); the strength of this decreased following adjustment for possible confounders ($p = 0.11$; table 4).

### 3.3. Behavioural measures

There was no evidence for between-group differences using the outfit choice task (test for trend $p = 0.24$), or in terms of the number of biscuits consumed (test for trend $p = 0.52$) (table 5).

### 3.4. Implicit measures

There was no clear statistical evidence of between-group differences in the word IAT (test for trend $p = 0.45$). Following adjustment for possible confounding variables in the images IAT, there was some evidence that participants were faster to respond to slightly overweight images paired with 'self' words following training with overweight images compared to other images (test for trend $p = 0.04$). There was no evidence for between-group differences in speed of reaction for 'beauty' minus 'ugly' words [5], or for 'positive' minus 'neutral' words. There was no evidence for between-group differences in LDT word recall (table 5).

**Table 5.** Results for behavioural and implicit tasks (see electronic supplementary material). Bold print indicates $p < 0.05$.

| | unadjusted comparison with those adapted to neither over- nor underweight images (Session 2 scores adjusted for Session 1 scores) | | | comparison with those adapted to neither over- nor underweight images, adjusted for number of images seen, time since last training, measured BMI, time spent using media ($N = 88$) | | |
| --- | --- | --- | --- | --- | --- | --- |
| | adapted to underweight images | adapted to overweight images | test of trend across groups | adapted to underweight images | adapted to overweight images | test of trend across groups |
| outfit choice task score ($N = 92$) | 0.06 (−0.14, 0.26), $p = 0.54$ | −0.11 (−0.30, 0.09), $p = 0.29$ | $p = 0.24$ | 0.05 (−0.16, 0.26), $p = 0.66$ ($N = 89$) | −0.12 (−0.34, 0.10), $p = 0.27$ ($N = 89$) | $p = 0.27$ |
| weight biscuits eaten ($N = 90$) | −3.30 (−9.35, 2.75), $p = 0.28$ | −2.68 (−8.80, 3.43), $p = 0.39$ | $p = 0.52$ | −4.34 (−10.62, 1.94), $p = 0.17$ ($N = 87$) | −2.90 (−9.40, 1.94), $p = 0.38$ ($N = 87$) | $p = 0.38$ |
| IAT words ($N = 91$) | −0.08 (−0.29, 0.12), $p = 0.42$ | −0.13 (−0.33, 0.08), $p = 0.22$ | $p = 0.45$ | −0.09 (−0.31, 0.13), $p = 0.43$ | −0.19 (−0.42, 0.03), $p = 0.09$ | $p = 0.24$ |
| IAT images ($N = 91$) | 0.01 (−0.14, 0.15), $p = 0.94$ | −0.12 (−0.26, 0.02), $p = 0.09$ | $p = 0.14$ | **−0.002 (−0.15, 0.15), $p = 0.97$** | **−0.17 (−0.33, −0.02), $p = 0.03$** | **$p = 0.04$** |
| LDT recall of beauty words (%) ($N = 90$) | 4.52 (−1.44, 10.47), $p = 0.14$ | −1.93 (−8.03, 4.18), $p = 0.53$ | $p = 0.09$ | 4.95 (−1.16, 11.06), $p = 0.11$ | −1.80 (−8.22, 4.61), $p = 0.58$ | $p = 0.07$ |
| LDT recall of ugly words (%) ($N = 90$) | 1.45 (−3.96, 6.85), $p = 0.60$ | 0.07 (−5.34, 5.47), $p = 0.98$ | $p = 0.83$ | 1.37 (−4.31, 7.05), $p = 0.63$ | −1.08 (−6.96, 4.80), $p = 0.72$ | $p = 0.69$ |

# 4. Discussion

Perception of the size of others' bodies changed according to the images shown during training, with those shown 'underweight' training images subsequently viewing fewer test images as underweight and more images as overweight, and vice versa. Avatar 'ideal' body also changed following training, with those shown underweight images creating a lower weight ideal than those shown overweight images. These results were in the expected direction.

However, contrary to our primary outcome measure hypotheses, there was no evidence of change on VAS measures of own size; there was also no change in satisfaction with own size. There were no between training group differences in questionnaire, implicit or behavioural measures of body dissatisfaction.

Participants created smaller 'own size' avatars after training with underweight as opposed to overweight images, and this was the opposite of what we had hypothesized. We believe that this is because we did not anticipate that the effects of the training on how participants viewed the starting size of the avatar would hugely outweigh any effects of the adaptation on their perception of their own size.

This study builds on previous work by ourselves and others aiming to understand the effect of exposure to images of women of different sizes on perception of and satisfaction with own size [12,16–18]. The study was appropriately powered to find between-group differences, and both subjects and experimenters were blind to randomization group. The stimuli used were carefully developed to be as ecologically valid as possible, and had also been rated by over 100 participants with similar characteristics as participants, to ensure they were viewed as being over, under or neither over- nor underweight. Although we used a range of outcome measures, we pre-registered our study and pre-defined our primary outcome of interest.

In terms of limitations, most participants were students at either the University of Oxford or Oxford Brookes University, and this may limit its generalizability to other populations. Recruitment of women from the 18 to 30 age group also limits generalizability outside this age group, and to men. We feel this is a good place to begin this work, as this group has high levels of body dissatisfaction and eating disorders, but it could be usefully repeated in men, and in a wider age group. Although we were not purely reliant on measures that could be biased by demand characteristics, some measures may be influenced in this way by subjects who had identified the purpose of the study. Using the same technique to measure perceived body size of both self and others would have made it possible to compare the two.

In terms of perception of the size of others' bodies, this study is consistent with previous results, which have used stretched photographs, line drawings and photographs [10–14,33,34]. Our study trained participants over a longer time scale, and one third of our participants had not done any training for more than six hours before the follow-up measurements, suggesting these effects on perception of others' bodies are quite long-lasting, which is consistent with our own previous work [16].

In terms of perception of own body, we did not replicate our previous finding, as we found no differences in VAS-rated perception of own size. However, using the avatar as a visual aid to allow participants to show us their perceived own size, we found that participants created a smaller 'self' avatar following exposure to underweight images.

In terms of satisfaction with own body size, we also found different results from our previous study [19]. We found no differences in VAS- or questionnaire-rated satisfaction with body size. These results fit with those described in another recent study [35]. In questionnaire measures of self-esteem and positive affect we found some suggestion that participants felt more positive in general after training with underweight bodies.

The study methodology does not allow definitive specification of the mechanism of action of change in perception of the size of others' bodies. However, many authors have argued that this change may be an 'adaptation effect' [10–12,15,36]. Adaptation occurs when a subject has a prolonged exposure to a stimulus. They then experience after-effects of this exposure when they subsequently view other, similar, stimuli. A classic example of this is of focusing on an object moving in one direction; stationary stimuli viewed immediately afterwards appear to be moving in the opposite direction [37]. Adaptation effects in relation to body size describe that repeated or extended exposure to images of smaller bodies leads to participants experiencing a perceptual after-effect of viewing subsequent images as larger, and vice versa. Such perceptual after-effects could plausibly be the underlying mechanism here.

A related interpretation follows from the 'norm-based' coding of bodies described by Rhodes and colleagues [38]; the central idea is that representations of bodies, like those of faces, are 'norm-based'. That is to say, each dimension is coded by pairs of neural channels tuned to opposite extremes, with

'normal' being represented by equal activation in both channels. Repeated exposure to bodies of different sizes leads to the body weight 'norm' shifting, so that what is perceived as 'normal' becomes more underweight following exposure to multiple underweight images, and more overweight following exposure to multiple overweight images.

With relation to our use of the avatar to demonstrate perception of own size, we believe that our hypothesis failed to take into account the effect of the training on participant perception of the starting size of the avatar. That is, according to body size adaptation effects as described above, the after-effect of exposure to smaller images would be that the avatar starting image would appear larger. Even if the participant viewed their own body as being larger in comparison to the images they had seen, the avatar image would appear larger too, and, in order to make the avatar better represent their own size, they would have to adjust it to make it smaller. We had not anticipated that the strong adaptation after-effect on the avatar size could outweigh any change in perceived own size in this way.

We have shown that repeated exposure, even for relatively brief periods of time (less than the average amount of time participants reported spending using social media, reading magazines or watching television), to images of women of different sizes changes perception of the size of other women, so that what is viewed as normal becomes smaller following exposure to images of smaller as opposed to larger women. This effect appears to also transfer so strongly to the perception of an avatar of a woman's body that we were unable to use the avatar method as we had planned, to measure changes in perceived own size. Repeated exposure to images of smaller 'normal' women also decreased the body size women viewed as 'ideal', as represented by the avatar.

This suggests that, at a population level, changing the size of bodies that people are regularly exposed to in the media would shift their perception of what constitutes a 'normal' and 'ideal' body size. It is not clear from this study that doing so would reliably effect satisfaction with own size. Similarly, using interventions like that used here in a population of people with a current eating disorder may also shift their perception of what body size is 'normal'; further research as to whether this would reduce their dissatisfaction with their own size would be informative. Some work suggests that perceived own size in those with an eating disorder is more easily shifted towards overweight [39].

Research ethics. The authors assert that all procedures contributing to this work comply with the ethical standards of the relevant national and institutional committees on human experimentation and with the Helsinki Declaration of 1975, as revised in 2008.

Data accessibility. The data forming the basis of the results presented here are available on the University of Bristol Repository: https://data.bris.ac.uk/data/dataset/my46h06p89rw163wu2du2jjzn [32].

Authors' contributions. H.B., C.J.H., I.P.-V., A.S., M.R.M, R.J.P. and M.R.B. conceived of and designed the study 2; K.N. and H.B. collected the data; H.B. conducted the statistical analysis; H.B. wrote the first draft of the manuscript; all authors helped with subsequent drafts. All authors gave final approval for publication.

Competing interests. We declare we have no competing interests.

Funding. This work was supported by a Wellcome Doctoral Training Fellowship (H.B.) at the University of Oxford, October 2014–July 2018, with funding from the NIHR Biomedical Research Centre, based at Oxford University Hospitals Trust, Oxford. M.R.M. is a member of the UK Centre for Tobacco Control Studies, a UKCRC Public Health Research: Centre of Excellence. Funding from British Heart Foundation, Cancer Research UK, Economic and Social Research Council, Medical Research Council and the National Institute for Health Research, under the auspices of the UK Clinical Research Collaboration, is gratefully acknowledged. M.R.M. and I.P.-V. are supported by the NIHR Bristol Health Biomedical Research Centre. C.J.H. is supported by the NIHR Oxford Health Biomedical Research Centre. The views expressed are those of the authors and not necessarily those of the NHS, the NIHR or the Department of Health.

Acknowledgements. We thank the study participants for their time.

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
