## [Reviewer comments · Royal Society Open Science]

Review History

RSOS-190704.R0 (Original submission)

Review form: Reviewer 1

Is the manuscript scientifically sound in its present form?

No

Are the interpretations and conclusions justified by the results?

No

Is the language acceptable?

Yes

Do you have any ethical concerns with this paper?

No

Have you any concerns about statistical analyses in this paper?

Yes

Recommendation?

Major revision is needed (please make suggestions in comments)

Comments to the Author(s)

This manuscript describes an experiment on the influence of adaptation to body stimuli on many responses, most importantly including the perception of their own and others' body size, and satisfaction with own size.

While the manuscript covers an interesting and topical issue, there are many problems that need to be addressed before publication can be recommended. These include factual inaccuracies and claims, inadequacies in detail and completeness of the methods and results sections, and major misinterpretation of the data in the discussion section.

PROBLEMS

The biggest problem with this manuscript is the interpretation of the avatar data.

The authors writes: "Some results were in the opposite direction to that hypothesised.

Participants created smaller "own size" avatars after training with underweight as opposed to overweight images, suggesting they viewed themselves as smaller following training with underweight images."

Hypotheses were not explicit stated, but I do not believe that this effect is "opposite" in direction to a sensible hypothesis. It is the same effect shown in references 8-10 and 13-15, amongst others.

Following adaptation to an underweight body, a conventional aftereffect would make stimuli appear larger. Hence, when the participant sees an avatar that (objectively) corresponding exactly to their size, it will appear too large. Participants would hence have to reduce its size so that it appears to them to be correct (matching own body). Hence a smaller avatar is selected.

From this misinterpretation, much of the discussion must be changed. Eg: "With relation to perception of own size, an adaptation explanation doesn't explain why women trained with "underweight" images in this study subsequently depicted themselves as smaller, despite rating more other women as overweight."

I would say that this is actually very consistent with adaptation explanation.

Also, see the following section:

"If her definition of "population-norm" is shifted to become smaller following exposure to "underweight" images (as it appears to be in this study), her perception of her own size, if it is defined in relation to the "population-norm", would move with it."

This reveals a misunderstanding of the detail of norm-based encoding, and its application to adaptation. In this model, the location of the body in "body space" (ref 31) does not moves with the norm - it is encoded by reference to the norm, and hence would appear further from it, i.e. aftereffect as shown in the results, but in the opposite direction to that described by the authors.

But given that the result itself has been misinterpreted, this section must change anyway (also the abstract).

It is not clear why were there such major differences in methods for own body vs. others.

Although effects are present for others pictures and own bodies (avatars, but not VAS), this makes it impossible to compare the size of the effects for these two, as was compared in ref 10.

This is a significant weakness in this paper.

A related issue: "To measure perceived body size in others, participants rated a series of 90 images of bodies(18), similar to those used in the training task,"

The details from the "training task" stimuli are unclear. How many training task images were chosen in total, and how many were in each category? Were they all 3 versions of the same individual, or only one of each? In general, similarity between stimuli (adaptation and test stimuli) produces larger aftereffects. Hence, there is higher likelihood of a significant effect for others (where adaptation and test stimuli are similar) compared to own body stimuli. With this in mind, it is essential to tell whether the adaptation and test stimuli are identical for "others" or whether they depicted different individuals. It is unclear currently.

In larger parts of this manuscript, the authors seem to be on a fishing expedition, throwing in as many measures as they possibly can, regardless of any hint of a hypothesis. This is a significant weakness for this paper. Some of these were repeated from previous paper (e.g. Maltesers replaced by oat cookies). I would advise the removal of many of the DVs.

Strangely the authors avoid the word "adaptation" particularly in the title and abstract but also much of the remaining. Instead the adaptation sessions are described as "training", which does not seem appropriate - there is no actual training. The authors also avoid the word adaptation in ref 16 (which may explain why I had not heard of this study - its absents may prevent interested researchers from finding the current manuscript too), but strangely it is used in the abstract which originally reported ref 16 (i.e. ref 29 - this ref should be removed, to avoid double citing/ gray literature).

There is no mention of satisfaction outcome in the abstract, despite that it is the focus of the study, according to the title. Yet it is not pre-registered as the primary outcome measure.

OTHER ISSUES

Remove p-values in abstract

"For example, more than one in three adults in the United States now meets criteria for obesity (6)."

Not all of the readers will reside in the USA. Do you have statistics for other countries, e.g. those in Europe or other industrialized countries?

"Limitations of these previous studies include small sample size (14, 15), and images lacking in ecological validity since they were uniformly modified to appear of different sizes rather than manipulated in line with typical patterns of weight gain (16)."

The author should be specific about the studies being criticize here. More specific, very few use stimuli that are uniformly modify (except ref 9), and many use actual photographs (11, 12) or stimuli that are "manipulated in line with typical patterns of weight gain" (e.g. 8, 13-15). Sample sizes used in refs 14 & 15 agree with the general methods in adaptation aftereffect research, and were large enough to find significant effects in each case, so they do not suffer inadequate power. See also the following text in the discussion: "The stimuli used were carefully developed to be more ecologically valid than those used previously."

"Perhaps the key limitation of previous studies is that they involved a single, brief exposure to the images."

Many of the cited studies use "top-up" adaptation. Again, please be specific about which studies you are criticizing.

"10-point VAS measure of own size from too thin (0) to too fat (10)"

Do you mean "too fat to be an accurate representation of my current body size", or "fatter than I'd like to be"? Was the participant aware of which response they should give? Also, please define "VAS" for the uninitiated reader.

"participants were also given two minutes to alter the weight, bust, waist and hip sizes of a computer avatar"

How were these 4 separate DVs processed/ which were analysed? Results section is also unclear.

"both of which had been created to be the same height as the participant: the first of these avatars began at their pre-reported body weight plus 10%"

Please include details of when was height and weight measured.

"participants were primed for the task by adjusting the Avatar to their own shape and size"

What does this mean?

“To prevent low level visual adaptation, the images were spatially jittered by an amount varying randomly from the centre of the screen by up to +/- 51 pixels in the x coordinate and +/- 38 pixels in the y coordinate (approximately 10% in each direction of the 1024 by 768 pixel display).” Given that eye movements were not constrained, how would variation of image position on screen serve to prevent low level adaptation? Note that ref 14 & 15 address this by changing body orientation (eye rotations are less likely than lateral eye movements), while Hummel et al 2012 (Perception) showed that low level adaptation of rectangles does not transfer to bodies. Also, recent study by Brookes et al (2018, Royal Society Open Science) showed that body adaptation generally has only a small low-level contribution. Also note that the variations of +/-51pix in 1024, or +/-38pix in 768 is approximately 5% in each direction (10% in total) for each dimension.

“These are presented at baseline for the entire sample, and by randomization group.”
Please state where?

“The effects of training on judgement of own size were assessed using linear regression to compare post-intervention scores, adjusted for pre-intervention scores.”
How was this adjustment made? (Same question for other dependent variables where a similar adjustment is mentioned.)

“Randomisation group was treated as a categorical measure throughout.”
Repeated from 2 sentences earlier.

“Analyses were repeated after adjusting for potentially confounding variables: number of images seen during training (as a measure of the amount of exposure participants had to the different images), time since last training [in tertiles] (in case any effects reduce over time), measured BMI (which influences perception of body size of others (18), and time spent using media [in tertiles] (which may negate or reduce the effect of the study images).”
Why was the number of images seen during training not constant for all observers?
Why was time since last training/time spent using media divided into tertiles (rather than being treated as continuous variable)? How was this done, and what were the cut-offs? What were the details of the media use questionnaire?

“Primary Outcome Measures: Own Size and Satisfaction with Own Size”
Insert “VAS” to distinguish this from avatar own size dependent variable.

“Explicit Measures”
The placement of this heading excludes the two previous sections (VAS own body size and satisfaction, and perception of others’ body size), yet in the methods section, these are deemed explicit measures.

“There were differences between training groups in the BMI’s of their “own size” avatars (test of trend $p=0.007$) and “ideal size” avatars (test of trend $p=0.03$)”

“The use of a range of outcome measures meant we were not reliant purely on explicit measures which could be affected by demand characteristics.”
This statement might carry some weight if all DVs – those susceptible to demand characteristics and those that are not – showed significant effects. Besides, it is unclear which DVs are immune to demand characteristics.

Reference #29

This is an abstract which appears to describe the same study as ref #16. Avoid citing the gray literature.

There are several relevant papers that have not been cited, such as a recent review of body adaptation plus several empirical papers
 Challinor et al., 2017 (International Journal of Medical Research),
 Hummel, Grabhorn, & Mohr, 2012 (Perception);
 Mohr, Rickmeyer, Hummel, Ernst, & Grabhorn, 2016 (Perception).

Review form: Reviewer 2

Is the manuscript scientifically sound in its present form?

Yes

Are the interpretations and conclusions justified by the results?

Yes

Is the language acceptable?

Yes

Do you have any ethical concerns with this paper?

No

Have you any concerns about statistical analyses in this paper?

No

Recommendation?

Accept with minor revision (please list in comments)

Comments to the Author(s)

The paper provides a useful contribution to the literature, and is written in a clear and comprehensive way which easily allows replication, and does a good job in attempting to understand its unexpected results. Some minor adjustments could be made prior to publication, listed below:

METHODS:

It appears that the paper utilised implicit measures alongside explicit measures to address bias. As such, it would be useful to include a statement regarding this in the Methods section to make clear how the experiment addressed potential sources of bias.

Page 10, paragraph 3: Please change "half way" to "halfway".

RESULTS:

If there was any missing data in the questionnaires, it would be helpful to include a statement for how this was addressed.

Page 18, lines 32,33, 37, & 37: There appears to be missing 2 references.

INTRODUCTION/DISCUSSION: It would be useful to emphasise the clinical relevance of the experiment, either in the introduction, discussion, or both. If viewing overweight/underweight bodies has an impact on our own body image, how might that be applied in a clinical context? The authors do state that interventions drawing on this research might have potential use in preventing/treating body dissatisfaction, but do not give any information as to what these interventions might look like. Emphasising the clinical relevance of the research will be useful for the reader to understand why this research and its findings are of importance.

Decision letter (RSOS-190704.R0)

22-Aug-2019

Dear Dr Bould,

The editors assigned to your paper ("Does repeatedly viewing overweight versus underweight images change satisfaction with own body size?") have now received comments from reviewers. We would like you to revise your paper in accordance with the referee and Associate Editor suggestions which can be found below (not including confidential reports to the Editor). Please note this decision does not guarantee eventual acceptance.

Please submit a copy of your revised paper before 14-Sep-2019. Please note that the revision deadline will expire at 00.00am on this date. If we do not hear from you within this time then it will be assumed that the paper has been withdrawn. In exceptional circumstances, extensions may be possible if agreed with the Editorial Office in advance. We do not allow multiple rounds of revision so we urge you to make every effort to fully address all of the comments at this stage. If deemed necessary by the Editors, your manuscript will be sent back to one or more of the original reviewers for assessment. If the original reviewers are not available, we may invite new reviewers.

- Data accessibility

<http://datadryad.org/submit?journalID=RSOS&manu=RSOS-190704>

- Competing interests

- Authors' contributions

- Acknowledgements

- Funding statement

on behalf of Dr Jonathan Roiser (Associate Editor) and Essi Viding (Subject Editor)
openscience@royalsociety.org

Reviewers' Comments to Author:

Reviewer: 1

This manuscript describes an experiment on the influence of adaptation to body stimuli on many responses, most importantly including the perception of their own and others' body size, and satisfaction with own size.

While the manuscript covers an interesting and topical issue, there are many problems that need to be addressed before publication can be recommended. These include factual inaccuracies and claims, inadequacies in detail and completeness of the methods and results sections, and major misinterpretation of the data in the discussion section.

PROBLEMS

The biggest problem with this manuscript is the interpretation of the avatar data.

The authors writes: "Some results were in the opposite direction to that hypothesised.

Participants created smaller "own size" avatars after training with underweight as opposed to overweight images, suggesting they viewed themselves as smaller following training with underweight images."

Hypotheses were not explicit stated, but I do not believe that this effect is "opposite" in direction to a sensible hypothesis. It is the same effect shown in references 8-10 and 13-15, amongst others.

Following adaptation to an underweight body, a conventional aftereffect would make stimuli appear larger. Hence, when the participant sees an avatar that (objectively) corresponding exactly to their size, it will appear too large. Participants would hence have to reduce its size so that it appears to them to be correct (matching own body). Hence a smaller avatar is selected.

From this misinterpretation, much of the discussion must be changed. Eg: "With relation to perception of own size, an adaptation explanation doesn't explain why women trained with "underweight" images in this study subsequently depicted themselves as smaller, despite rating more other women as overweight."

I would say that this is actually very consistent with adaptation explanation.

Also, see the following section:

"If her definition of "population-norm" is shifted to become smaller following exposure to "underweight" images (as it appears to be in this study), her perception of her own size, if it is defined in relation to the "population-norm", would move with it."

This reveals a misunderstanding of the detail of norm-based encoding, and its application to adaptation. In this model, the location of the body in "body space" (ref 31) does not moves with the norm - it is encoded by reference to the norm, and hence would appear further from it, i.e. aftereffect as shown in the results, but in the opposite direction to that described by the authors. But given that the result itself has been misinterpreted, this section must change anyway (also the abstract).

It is not clear why were there such major differences in methods for own body vs. others.

Although effects are present for others pictures and own bodies (avatars, but not VAS), this makes it impossible to compare the size of the effects for these two, as was compared in ref 10. This is a significant weakness in this paper.

A related issue: "To measure perceived body size in others, participants rated a series of 90 images of bodies(18), similar to those used in the training task,"

The details from the "training task" stimuli are unclear. How many training task images were chosen in total, and how many were in each category? Were they all 3 versions of the same individual, or only one of each? In general, similarity between stimuli (adaptation and test stimuli) produces larger aftereffects. Hence, there is higher likelihood of a significant effect for others (where adaptation and test stimuli are similar) compared to own body stimuli. With this in mind, it is essential to tell whether the adaptation and test stimuli are identical for "others" or whether they depicted different individuals. It is unclear currently.

In larger parts of this manuscript, the authors seem to be on a fishing expedition, throwing in as many measures as they possibly can, regardless of any hint of a hypothesis. This is a significant weakness for this paper. Some of these were repeated from previous paper (e.g. Maltesers replaced by oat cookies). I would advise the removal of many of the DVs.

Strangely the authors avoid the word "adaptation" particularly in the title and abstract but also much of the remaining. Instead the adaptation sessions are described as "training", which does not seem appropriate - there is no actual training. The authors also avoid the word adaptation in ref 16 (which may explain why I had not heard of this study - its absents may prevent interested researchers from finding the current manuscript too), but strangely it is used in the abstract which originally reported ref 16 (i.e. ref 29 - this ref should be removed, to avoid double citing/ gray literature).

There is no mention of satisfaction outcome in the abstract, despite that it is the focus of the study, according to the title. Yet it is not pre-registered as the primary outcome measure.

OTHER ISSUES

Remove p-values in abstract

“For example, more than one in three adults in the United States now meets criteria for obesity (6).”

Not all of the readers will reside in the USA. Do you have statistics for other countries, e.g. those in Europe or other industrialized countries?

“Limitations of these previous studies include small sample size (14, 15), and images lacking in ecological validity since they were uniformly modified to appear of different sizes rather than manipulated in line with typical patterns of weight gain (16).”

The author should be specific about the studies being criticize here. More specific, very few use stimuli that are uniformly modify (except ref 9), and many use actual photographs (11, 12) or stimuli that are “manipulated in line with typical patterns of weight gain” (e.g. 8, 13-15). Sample sizes used in refs 14 & 15 agree with the general methods in adaptation aftereffect research, and were large enough to find significant effects in each case, so they do not suffer inadequate power. See also the following text in the discussion: “The stimuli used were carefully developed to be more ecologically valid than those used previously.”

“Perhaps the key limitation of previous studies is that they involved a single, brief exposure to the images.”

Many of the cited studies use “top-up” adaptation. Again, please be specific about which studies you are criticizing.

“10-point VAS measure of own size from too thin (0) to too fat (10)”

Do you mean “too fat to be an accurate representation of my current body size”, or “fatter than I’d like to be”? Was the participant aware of which response they should give? Also, please define “VAS” for the uninitiated reader.

“participants were also given two minutes to alter the weight, bust, waist and hip sizes of a computer avatar”

How were these 4 separate DVs processed/which were analysed? Results section is also unclear.

“both of which had been created to be the same height as the participant: the first of these avatars began at their pre-reported body weight plus 10%”

Please include details of when was height and weight measured.

“participants were primed for the task by adjusting the Avatar to their own shape and size”
What does this mean?

“To prevent low level visual adaptation, the images were spatially jittered by an amount varying randomly from the centre of the screen by up to +/- 51 pixels in the x coordinate and +/- 38 pixels in the y coordinate (approximately 10% in each direction of the 1024 by 768 pixel display).” Given that eye movements were not constrained, how would variation of image position on screen serve to prevent low level adaptation? Note that ref 14 & 15 address this by changing body orientation (eye rotations are less likely than lateral eye movements), while Hummel et al 2012 (Perception) showed that low level adaptation of rectangles does not transfer to bodies. Also, recent study by Brookes et al (2018, Royal Society Open Science) showed that body adaptation generally has only a small low-level contribution.

Also note that the variations of +/-51pix in 1024, or +/-38pix in 768 is approximately 5% in each direction (10% in total) for each dimension.

“These are presented at baseline for the entire sample, and by randomization group.”
Please state where?

“The effects of training on judgement of own size were assessed using linear regression to compare post-intervention scores, adjusted for pre-intervention scores.”
How was this adjustment made? (Same question for other dependent variables where a similar adjustment is mentioned.)

“Randomisation group was treated as a categorical measure throughout.”
Repeated from 2 sentences earlier.

“Analyses were repeated after adjusting for potentially confounding variables: number of images seen during training (as a measure of the amount of exposure participants had to the different images), time since last training [in tertiles] (in case any effects reduce over time), measured BMI (which influences perception of body size of others (18), and time spent using media [in tertiles] (which may negate or reduce the effect of the study images).”
Why was the number of images seen during training not constant for all observers?
Why was time since last training/time spent using media divided into tertiles (rather than being treated as continuous variable)? How was this done, and what were the cut-offs? What were the details of the media use questionnaire?

“Primary Outcome Measures: Own Size and Satisfaction with Own Size”
Insert “VAS” to distinguish this from avatar own size dependent variable.

“Explicit Measures”
The placement of this heading excludes the two previous sections (VAS own body size and satisfaction, and perception of others’ body size), yet in the methods section, these are deemed explicit measures.

“There were differences between training groups in the BMI’s of their “own size” avatars (test of trend $p=0.007$) and “ideal size” avatars (test of trend $p=0.03$)”

“The use of a range of outcome measures meant we were not reliant purely on explicit measures which could be affected by demand characteristics.”
This statement might carry some weight if all DVs – those susceptible to demand characteristics and those that are not – showed significant effects. Besides, it is unclear which DVs are immune to demand characteristics.

Reference #29

This is an abstract which appears to describe the same study as ref #16. Avoid citing the gray literature.

There are several relevant papers that have not been cited, such as a recent review of body adaptation plus several empirical papers
Challinor et al., 2017 (International Journal of Medical Research),
Hummel, Grabhorn, & Mohr, 2012 (Perception);
Mohr, Rickmeyer, Hummel, Ernst, & Grabhorn, 2016 (Perception).

Reviewer: 2

Comments to the Author(s)

The paper provides a useful contribution to the literature, and is written in a clear and comprehensive way which easily allows replication, and does a good job in attempting to

understand its unexpected results. Some minor adjustments could be made prior to publication, listed below:

METHODS:

It appears that the paper utilised implicit measures alongside explicit measures to address bias. As such, it would be useful to include a statement regarding this in the Methods section to make clear how the experiment addressed potential sources of bias.

Page 10, paragraph 3: Please change "half way" to "halfway".

RESULTS:

If there was any missing data in the questionnaires, it would be helpful to include a statement for how this was addressed.

Page 18, lines 32,33, 37, & 37: There appears to be missing 2 references.

INTRODUCTION/DISCUSSION: It would be useful to emphasise the clinical relevance of the experiment, either in the introduction, discussion, or both. If viewing overweight/underweight bodies has an impact on our own body image, how might that be applied in a clinical context? The authors do state that interventions drawing on this research might have potential use in preventing/treating body dissatisfaction, but do not give any information as to what these interventions might look like. Emphasising the clinical relevance of the research will be useful for the reader to understand why this research and its findings are of importance.

Author's Response to Decision Letter for (RSOS-190704.R0)

See Appendix A.

RSOS-190704.R1 (Revision)

Review form: Reviewer 1

Is the manuscript scientifically sound in its present form?

No

Are the interpretations and conclusions justified by the results?

No

Is the language acceptable?

Yes

Do you have any ethical concerns with this paper?

No

Have you any concerns about statistical analyses in this paper?

Yes

Recommendation?

Reject

Comments to the Author(s)

I was disappointed to read the revision and associated response to referees, as in many cases the authors have not follow to my suggestions, but instead argue against. In majority all cases, I find their arguments unconvincing, and the revision to be only a small improvement on some superficial issues. Many of the major failings of this manuscript remain. In particular, the hypotheses, which the authors acknowledge are flawed, are not consistent to the literature that are claimed to motivate them (e.g. refs 10-18).

Review form: Reviewer 2

Is the manuscript scientifically sound in its present form?

Yes

Are the interpretations and conclusions justified by the results?

Yes

Is the language acceptable?

Yes

Do you have any ethical concerns with this paper?

No

Have you any concerns about statistical analyses in this paper?

Yes

Recommendation?

Accept as is

Comments to the Author(s)

The author(s) have done a good job in addressing my previous comments.

Review form: Reviewer 3

Is the manuscript scientifically sound in its present form?

Yes

Are the interpretations and conclusions justified by the results?

Yes

Is the language acceptable?

Yes

Do you have any ethical concerns with this paper?

No

Have you any concerns about statistical analyses in this paper?

No

Recommendation?

Accept with minor revision (please list in comments)

Comments to the Author(s)

The manuscript describes a study looking at the effects of repeated exposure to under, over and 'normal' weight bodies on perception of own and other body size, and also own body satisfaction. The authors employ an elegant study design, with participants randomised to one of three groups and then undergoing a 7 days worth of training with different sized body images. Measures of others body size, own body size and body satisfaction were made pre- and post-training in two lab visits

The hypotheses and primary outcome measures were pre-registered, which is a strength.

There were no differences between the groups in bodily satisfaction - the primary outcome measure. The tasks used to measure changes in others vs own body size differed in their methodology which complicates the findings pertaining to perceptual shifts.

For others body size participants were presented with 90 images of women's bodies and asked to make explicit rating of whether each was under, over or normal weight. Using this task they showed that following training with underweight images, participants rated more images as overweight (and fewer as underweight). This result was mirrored following training with overweight images. This is consistent with the authors hypotheses, that training with over or underweight images will shift perception of body size in the opposite direction.

For perception of own body size participants were instructed to adjust an avatar until it matched their own body size. Here they found that training with underweight images made participants adjust their avatar to appear smaller (and vice versa for training with overweight images). While seemingly inconsistent with their hypothesis, and the findings for the perception of others bodies, in the revised version of the manuscript that I am reviewing the authors discuss the likely possibility that adaptation has shifted the initial perception of the avatar (to appear larger in the case of underweight training) and so participants adjust the avatar smaller to compensate for this. In fact, considering the differences in measurement across own and other body perception, long-term visual adaptation could plausibly account for both of the effects measured.

This is actually very interesting and adds to a broad literature suggesting that adaptation aftereffects can persist for many days and can build up outside of standard laboratory settings (Carbon & Ditye, *Frontiers in Psychology*, 2012).

It is, in my view, permissible to keep the hypotheses laid out in the introduction as they are (i.e. not accounting for the effects of adaption in the initial perception of the avatar) and to describe the interesting perceptual shift explanation for these results in the discussion, as the authors have done. My only suggestion to improve the manuscript is to carefully re-read and then re-structure the last part of the discussion in this regard, as the writing here could be clearer and these findings could be contextualised more in light of the broader literature on body size adaptation (i.e. perhaps explain what an adaptation aftereffect is and take a sentence or two to explain some of the common findings in relation to body size adaptation).

Decision letter (RSOS-190704.R1)

14-Jan-2020

Dear Dr Bould,

On behalf of the Editors, I am pleased to inform you that your Manuscript RSOS-190704.R1 entitled "Does repeatedly viewing overweight versus underweight images change perception of satisfaction with own body size?" has been accepted for publication in Royal Society Open

Science subject to minor revision in accordance with the referee suggestions. Please find the referees' comments at the end of this email.

The reviewers and Subject Editor have recommended publication, but also suggest some minor revisions to your manuscript. Therefore, I invite you to respond to the comments and revise your manuscript.

- Ethics statement

- Data accessibility

If you wish to submit your supporting data or code to Dryad (<http://datadryad.org/>), or modify your current submission to dryad, please use the following link:
<http://datadryad.org/submit?journalID=RSOS&manu=RSOS-190704.R1>

- Competing interests

- Authors' contributions

- Acknowledgements

- Funding statement

Because the schedule for publication is very tight, it is a condition of publication that you submit the revised version of your manuscript before 23-Jan-2020. Please note that the revision deadline will expire at 00.00am on this date. If you do not think you will be able to meet this date please let me know immediately.

Kind regards,

Lianne Parkjhouse
Editorial Coordinator
Royal Society Open Science
openscience@royalsociety.org

on behalf of Dr Jonathan Roiser (Associate Editor) and Professor Essi Viding (Subject Editor)
openscience@royalsociety.org

Associate Editor Comments to Author (Dr Jonathan Roiser):

One of the reviewers was satisfied with the revised version, but the other was not and was not supportive of publication. Therefore we approached a third reviewer whose comments are appended below. This new reviewer feels that the manuscript is almost ready for publication, but that the Discussion section requires some re-writing in order to clarify the interpretation of the results for the reader. We feel that you should be able to address this without much difficulty, and therefore we are issuing a minor revision decision. We do not anticipate that the manuscript will need to be sent out again for review.

Reviewer 3

The manuscript describes a study looking at the effects of repeated exposure to under, over and 'normal' weight bodies on perception of own and other body size, and also own body satisfaction. The authors employ an elegant study design, with participants randomised to one of three groups and then undergoing a 7 days worth of training with different sized body images. Measures of others body size, own body size and body satisfaction were made pre- and post-training in two lab visits

The hypotheses and primary outcome measures were pre-registered, which is a strength.

There were no differences between the groups in bodily satisfaction - the primary outcome measure. The tasks used to measure changes in others vs own body size differed in their methodology which complicates the findings pertaining to perceptual shifts.

For others body size participants were presented with 90 images of women's bodies and asked to make explicit rating of whether each was under, over or normal weight. Using this task they showed that following training with underweight images, participants rated more images as overweight (and fewer as underweight). This result was mirrored following training with overweight images. This is consistent with the authors hypotheses, that training with over or underweight images will shift perception of body size in the opposite direction.

For perception of own body size participants were instructed to adjust an avatar until it matched their own body size. Here they found that training with underweight images made participants adjust their avatar to appear smaller (and vice versa for training with overweight images). While seemingly inconsistent with their hypothesis, and the findings for the perception of others bodies, in the revised version of the manuscript that I am reviewing the authors discuss the likely possibility that adaptation has shifted the initial perception of the avatar (to appear larger in the case of underweight training) and so participants adjust the avatar smaller to compensate for this. In fact, considering the differences in measurement across own and other body perception, long-term visual adaptation could plausibly account for both of the effects measured.

This is actually very interesting and adds to a broad literature suggesting that adaptation aftereffects can persist for many days and can build up outside of standard laboratory settings (Carbon & Ditye, *Frontiers in Psychology*, 2012).

It is, in my view, permissible to keep the hypotheses laid out in the introduction as they are (i.e. not accounting for the effects of adaption in the initial perception of the avatar) and to describe the interesting perceptual shift explanation for these results in the discussion, as the authors have done. My only suggestion to improve the manuscript is to carefully re-read and then re-structure the last part of the discussion in this regard, as the writing here could be clearer and these

findings could be contextualised more in light of the broader literature on body size adaptation (i.e. perhaps explain what an adaptation aftereffect is and take a sentence or two to explain some of the common findings in relation to body size adaptation).

Reviewer comments to Author:

Reviewer: 1

Comments to the Author(s)

I was disappointed to read the revision and associated response to referees, as in many cases the authors have not follow to my suggestions, but instead argue against. In majority all cases, I find their arguments unconvincing, and the revision to be only a small improvement on some superficial issues. Many of the major failings of this manuscript remain. In particular, the hypotheses, which the authors acknowledge are flawed, are not consistent to the literature that are claimed to motivate them (e.g. refs 10-18).

Reviewer: 2

Comments to the Author(s)

The author(s) have done a good job in addressing my previous comments.

Reviewer: 3

Comments to the Author(s)

The manuscript describes a study looking at the effects of repeated exposure to under, over and 'normal' weight bodies on perception of own and other body size, and also own body satisfaction. The authors employ an elegant study design, with participants randomised to one of three groups and then undergoing a 7 days worth of training with different sized body images. Measures of others body size, own body size and body satisfaction were made pre- and post-training in two lab visits

The hypotheses and primary outcome measures were pre-registered, which is a strength.

There were no differences between the groups in bodily satisfaction - the primary outcome measure. The tasks used to measure changes in others vs own body size differed in their methodology which complicates the findings pertaining to perceptual shifts.

For others body size participants were presented with 90 images of women's bodies and asked to make explicit rating of whether each was under, over or normal weight. Using this task they showed that following training with underweight images, participants rated more images as overweight (and fewer as underweight). This result was mirrored following training with overweight images. This is consistent with the authors hypotheses, that training with over or underweight images will shift perception of body size in the opposite direction.

For perception of own body size participants were instructed to adjust an avatar until it matched their own body size. Here they found that training with underweight images made participants adjust their avatar to appear smaller (and vice versa for training with overweight images). While seemingly inconsistent with their hypothesis, and the findings for the perception of others bodies, in the revised version of the manuscript that I am reviewing the authors discuss the likely possibility that adaptation has shifted the initial perception of the avatar (to appear larger in the case of underweight training) and so participants adjust the avatar smaller to compensate for this. In fact, considering the differences in measurement across own and other body perception, long-term visual adaptation could plausibly account for both of the effects measured.

This is actually very interesting and adds to a broad literature suggesting that adaptation

aftereffects can persist for many days and can build up outside of standard laboratory settings (Carbon & Ditye, *Frontiers in Psychology*, 2012).

It is, in my view, permissible to keep the hypotheses laid out in the introduction as they are (i.e. not accounting for the effects of adaption in the initial perception of the avatar) and to describe the interesting perceptual shift explanation for these results in the discussion, as the authors have done. My only suggestion to improve the manuscript is to carefully re-read and then re-structure the last part of the discussion in this regard, as the writing here could be clearer and these findings could be contextualised more in light of the broader literature on body size adaptation (i.e. perhaps explain what an adaptation aftereffect is and take a sentence or two to explain some of the common findings in relation to body size adaptation).

Author's Response to Decision Letter for (RSOS-190704.R1)

See Appendix B.

Decision letter (RSOS-190704.R2)

03-Feb-2020

Dear Dr Bould,

It is a pleasure to accept your manuscript entitled "Does repeatedly viewing overweight versus underweight images change perception of satisfaction with own body size?" in its current form for publication in Royal Society Open Science. The comments of the reviewer(s) who reviewed your manuscript are included at the foot of this letter.

Kind regards,
Andrew Dunn
Royal Society Open Science Editorial Office

on behalf of Dr Jonathan Roiser (Associate Editor) and Essi Viding (Subject Editor)
openscience@royalsociety.org

Appendix A

Centre for Academic Mental Health

Population Health Science

Oakfield House, Oakfield Grove

University of Bristol

1.10.19

Dear Alice,

Re: Manuscript ID RSOS-190704 ("Does repeatedly viewing overweight versus underweight images change satisfaction with own body size?")

Many thanks for giving us the opportunity to respond to the reviewers' comments on our paper, which we have done so thoroughly below. We very much hope that RSOS will feel that the paper is now ready for publication.

Reviewers' Comments to Author:

Reviewer: 1

This manuscript describes an experiment on the influence of adaptation to body stimuli on many responses, most importantly including the perception of their own and others' body size, and satisfaction with own size.

While the manuscript covers an interesting and topical issue, there are many problems that need to be addressed before publication can be recommended.

PROBLEMS

1. The biggest problem with this manuscript is the interpretation of the avatar data.

The authors writes: "Some results were in the opposite direction to that hypothesised. Participants created smaller "own size" avatars after training with underweight as opposed to overweight images, suggesting they viewed themselves as smaller following training with underweight images." Hypotheses were not explicit stated, but I do not believe that this effect is "opposite" in direction to a sensible hypothesis. It is the same effect shown in references 8-10 and 13-15, amongst others. Following adaptation to an underweight body, a conventional aftereffect would make stimuli appear larger. Hence, when the participant sees an avatar that (objectively) corresponding exactly to their size, it will appear too large. Participants would hence have to reduce its size so that it appears to them to be correct (matching own body). Hence a smaller avatar is selected.

From this misinterpretation, much of the discussion must be changed. Eg: "With relation to perception of own size, an adaptation explanation doesn't explain why women trained with "underweight" images in this study subsequently depicted themselves as smaller, despite rating more other women as overweight."

I would say that this is actually very consistent with adaptation explanation.

Authors' response: Thank you for this insightful comment on our work. We had designed the Avatar task with the idea that it would capture subjects' perception of their own size, which we anticipated that they would judge in relation to the images they had seen (i.e. if adapted to small images, would feel larger in comparison and make self-avatar larger as a result). However, we failed to take into account the effects of the adaptation on how participants viewed the avatar itself, and that such effects would completely outweigh any change in perceived own size.

We have kept our introduction as previously written, as this was our aim and hypothesis, and have rewritten the discussion and conclusions substantially as follows, to make clear our error and better explain the results we found, as follows:

P21: "Participants created smaller "own size" avatars after training with underweight as opposed to overweight images, and this was the opposite of what we had hypothesised. We believe that this is because we did not anticipate that the effects of the training on how participants viewed the starting size of the avatar would hugely outweigh any effects of the adaptation on their perception of their own size."

P24: "With relation to our use of the Avatar to demonstrate perception of own size, we believe that our hypothesis was incorrect as we failed to take into account the effect of the training on participant perception of the starting size of the Avatar. That is, following exposure to smaller images, the Avatar image would appear larger. Even if the participant felt that their own body was also larger in comparison to the images they had seen, the Avatar image would appear larger too and therefore they would adjust it smaller to better represent their own size. We had not anticipated that the strong adaptation effect on the Avatar size would outweigh any change in perceived own size in this way."

P25: "We have shown that repeated exposure, even for relatively brief periods of time (less than the average amount of time participants reported spending using social media, reading magazines or watching television), to images of women of different sizes changes perception of the size of other women, so that what is viewed as "normal" becomes smaller following exposure to images of larger as opposed to smaller women. This effect appears to also transfer so strongly to the perception of an Avatar of a woman's body that we were unable to use the Avatar method as we had planned, to measure changes in perceived own size. Repeated exposure to images of smaller "normal" women also decreased the body size women viewed as "ideal", as represented by the Avatar."

We have also rewritten the Abstract as follows:

"Avatar-constructed ideal ($p=0.03$) and avatar-constructed perceived own body size ($p=0.007$) both decreased following exposure to underweight women, possibly due to adaptation affecting how the avatar was perceived. Repeated exposure to different sized bodies changes perception of the size of others' bodies, but we did not find evidence that it changes perceived own size."

2. Also, see the following section:

"If her definition of "population-norm" is shifted to become smaller following exposure to "underweight" images (as it appears to be in this study), her perception of her own size, if it is defined in relation to the "population-norm", would move with it."

This reveals a misunderstanding of the detail of norm-based encoding, and its application to

adaptation. In this model, the location of the body in “body space” (ref 31) does not moves with the norm – it is encoded by reference to the norm, and hence would appear further from it, i.e. aftereffect as shown in the results, but in the opposite direction to that described by the authors. But given that the result itself has been misinterpreted, this section must change anyway (also the abstract).

Authors’ response: We have changed the Discussion considerably in response to point (1) and this section has been removed.

3. It is not clear why were there such major differences in methods for own body vs. others. Although effects are present for others pictures and own bodies (avatars, but not VAS), this makes it impossible to compare the size of the effects for these two, as was compared in ref 10. This is a significant weakness in this paper.

Authors’ response: We appreciate with hindsight that aligning methods of comparing own and others’ bodies would have been helpful to enable comparison of the magnitude of the effects found, and we have added this to the Limitations section as follows:

P22: “Using the same technique to measure perceived body size of both self and others would have made it possible to compare effect sizes between the two.”

4. A related issue: “To measure perceived body size in others, participants rated a series of 90 images of bodies(18), similar to those used in the training task,”

The details from the “training task” stimuli are unclear. How many training task images were chosen in total, and how many were in each category? Were they all 3 versions of the same individual, or only one of each? In general, similarity between stimuli (adaptation and test stimuli) produces larger aftereffects. Hence, there is higher likelihood of a significant effect for others (where adaptation and test stimuli are similar) compared to own body stimuli. With this in mind, it is essential to tell whether the adaptation and test stimuli are identical for “others” or whether they depicted different individuals. It is unclear currently.

Authors’ response: Thank you for highlighting this lack of clarity. We have updated the “Training Task Stimuli” section as follows:

P12: “Eight images were chosen for each randomisation group (“underweight”, “neither over- nor underweight”, and “overweight”), and the images in the three groups were of the same individual women, but modified to appear of different weights. The eight images for the “neither over nor underweight” group were chosen by selecting those versions of the images which most online participants rated as neither over- nor underweight. The eight images for the underweight group were chosen by selecting images that around half the online participants rated as underweight, and the eight for the overweight group from those that around half the online participants had rated as overweight.

We have updated the Explicit Methods section as follows:

P8: "Twenty-four of these images were used in the training task itself (eight in each randomisation group), so each individual participant saw eight of these images during their training."

5. In larger parts of this manuscript, the authors seem to be on a fishing expedition, throwing in as many measures as they possibly can, regardless of any hint of a hypothesis. This is a significant weakness for this paper. Some of these were repeated from previous paper (e.g. Maltesers replaced by oat cookies). I would advise the removal of many of the DVs.

Authors' response: We disagree with the idea that we were on a "fishing expedition" – we pre-registered the study (<https://osf.io/pwd36/>), and clearly indicated our primary outcome measure and hypotheses. The aim of including a range of measures was to assess whether changes were observed across a range of measures, including those less susceptible to bias. Our purpose in including the range of DV's studied in the paper is to be entirely transparent to the reader that we studied a range of secondary measures, thus setting in context the findings seen with the avatar.

6. Strangely the authors avoid the word "adaptation" particularly in the title and abstract but also much of the remaining. Instead the adaptation sessions are described as "training", which does not seem appropriate - there is no actual training. The authors also avoid the word adaptation in ref 16 (which may explain why I had not heard of this study – its absents may prevent interested researchers from finding the current manuscript too), but strangely it is used in the abstract which originally reported ref 16 (i.e. ref 29 – this ref should be removed, to avoid double citing/gray literature).

Authors' response: Firstly, we apologise for the inadvertent double/gray literature citing and have removed this reference. Secondly, the word "adaptation" featured in early drafts of our previous paper, but we removed the term from the title and abstract of the final peer-reviewed article in response to reviewer comments that we did not know for certain that the underlying process leading to our results was adaptation. We therefore changed the title to discuss "exposure", and described the process of exposure to the images using the more neutral term "training".

Continuing to use the more neutral term "training" in this paper enables us to be consistent, as well as acknowledging that the paper doesn't seek to prove that the mechanism involved is adaptation. It also seems to us to be an appropriate way of concisely describing the process of going through the images, particularly for the participants, who understood that the purpose of the "training" was for them to get better at identifying whether each image they saw was the same as the previous one.

However, we take on board your concern that this may make the article harder to find, and will ask that the term "adaptation" be added to the article key words/search terms to guard against this.

7. There is no mention of satisfaction outcome in the abstract, despite that it is the focus of the study, according to the title. Yet it is not pre-registered as the primary outcome measure.

Authors' response: Thank you for highlighting this disparity. Satisfaction with size was not the primary outcome, but was nonetheless an important outcome. We have addressed your point both by updating the title to

“Does repeatedly viewing overweight versus underweight images change perception of and satisfaction with own body size?”

and amending the Abstract to:

“There was no evidence for a difference in our primary outcome measure (Visual Analogue Scale own size) or in satisfaction with own size.”

OTHER ISSUES

1. Remove p-values in abstract

Authors’ response: We are happy to keep or leave these according to the editor’s opinion.

2. “For example, more than one in three adults in the United States now meets criteria for obesity (6).”

Not all of the readers will reside in the USA. Do you have statistics for other countries, e.g. those in Europe or other industrialized countries?

Authors’ response: We have added the following

P3: “obesity rates are increasing globally (6). More than one in three adults in the United States (7) and 29% of adults in England (8) now meet criteria for obesity.”

This has added the following references: 6. Collaboration NCDRF. Trends in adult body-mass index in 200 countries from 1975 to 2014: a pooled analysis of 1698 population-based measurement studies with 19.2 million participants. Lancet. 2016;387(10026):1377-96 & 8. Digital N. Statistics on Obesity, Physical Activity and Diet, England. In: Digital N, editor.: NHS Digital; 2019).

3. “Limitations of these previous studies include small sample size (14, 15), and images lacking in ecological validity since they were uniformly modified to appear of different sizes rather than manipulated in line with typical patterns of weight gain (16).”

The author should be specific about the studies being criticize here. More specific, very few use stimuli that are uniformly modify (except ref 9), and many use actual photographs (11, 12) or stimuli that are “manipulated in line with typical patterns of weight gain” (e.g. 8, 13-15). Sample sizes used in refs 14 & 15 agree with the general methods in adaptation aftereffect research, and were large enough to find significant effects in each case, so they do not suffer inadequate power. See also the following text in the discussion: “The stimuli used were carefully developed to be more ecologically valid than those used previously.”

Authors’ response: Our discussion of the limitations of previous studies has focused on those which investigating whether such interventions can transfer from images of others to perceived own size, which is why we focus on references 14, 15 and 16 (our own previous study) in this section. We do

not feel that it is unreasonable to state that the sample size in the previous studies with sample sizes of 14 and 16 was small, and that this is a limitation. There is evidence from multiple fields that many studies are underpowered, and the resulting effect estimates may be imprecise and, because of the application of a p-value filter for claiming discovery, inflated (e.g. Button 2013 “Power failure: why small sample size undermines the reliability of neuroscience”). Hence it is important to replicate results in larger samples. We criticise our own previous work when we point out that our last study did not manipulate images in line with typical patterns of weight gain.

We have amended our discussion text as follows:

P22: “The stimuli used were carefully developed to be as ecologically valid as possible.”

4. *“Perhaps the key limitation of previous studies is that they involved a single, brief exposure to the images.”*

Many of the cited studies use “top-up” adaptation. Again, please be specific about which studies you are criticizing.

Authors’ response: On reflection, we think that describing the use of a single session of exposure as a “limitation” is the wrong word; it is simply that the focus of our study was to answer a different question than that addressed in previous studies: about the effect of repeated exposure over the course of a week. We have therefore rewritten the introduction as follows:

P5: “The key difference between previous studies and the study reported here, is that previous studies have involved a single session of exposure to the images, whereas our study involves repeated sessions of exposure over the course of a week. One would anticipate that any proposed intervention to address body dissatisfaction would need to be given repeatedly, in order to counteract ongoing exposure to underweight images in visual media; relatedly, in order to understand the effects of exposure to media images, we need to study the effects of repeated, as well as one-off, exposure to images of others’ bodies.

“The current study was designed to test the effect of repeated exposure (five minutes twice a day) over the course of a week to images of women of different sizes on women’s perception of their own and others’ size, and satisfaction with own size.”

5. *“10-point VAS measure of own size from too thin (0) to too fat (10)”*

Do you mean “too fat to be an accurate representation of my current body size”, or “fatter than I’d like to be”? Was the participant aware of which response they should give? Also, please define “VAS” for the uninitiated reader.

Authors’ response: Thank you for highlighting that this was not clear. We have amended the Methods as follows to address this point:

P8: “The pre-specified primary outcome measure was score on a 10-point VAS measure of own size from too thin (0) to too fat (10) (in response to the question “Please indicate on this scale what you feel is the size of your body at the moment”), and the secondary outcome measure was VAS satisfaction with own size from very dissatisfied (0) to very satisfied (10), in response to the

question “Please indicate on this scale how satisfied you feel about the size of your body at the moment”.

6. “participants were also given two minutes to alter the weight, bust, waist and hip sizes of a computer avatar”

How were these 4 separate DVs processed/which were analysed? Results section is also unclear.

Authors’ response: The DV’s for bust, waist and hips were not analysed separately, they were simply used to enable the participant to generate an accurate self-representation. We have added the following to the Methods section to clarify this:

P9: “The results were used to generate “Own size” Avatar Body Mass Index (BMI) and “Ideal Own size” Avatar BMI measures.”

We have also amended the Results subtitle to “Own and Ideal Size Avatar BMI”.

7. “both of which had been created to be the same height as the participant: the first of these avatars began at their pre-reported body weight plus 10%”

Please include details of when was height and weight measured.

Authors’ response: Thank you for spotting that we had not included this detail. We have added the following to the inclusive criteria:

P7: “have a Body Mass Index (BMI) within the healthy range of 18-25 kg/m², calculated from their self-reported weight and height”

the following to the “Explicit Measures” section:

P9: “We used the weight and height measurements reported in the screening questionnaire”

and the following to the end of the Explicit Measures section

P10: “At the end of the follow-up session, participants were weighed and their height was measured.”

8. “participants were primed for the task by adjusting the Avatar to their own shape and size”
What does this mean?

Authors’ response: We have rewritten this for clarity as follows:

P11: “Participants completed this task immediately after adjusting the Avatar to their own shape and size, with the intent that this would mean that they were primed to be thinking about their own body during the LDT.”

9. “To prevent low level visual adaptation, the images were spatially jittered by an amount varying randomly from the centre of the screen by up to +/- 51 pixels in the x coordinate and +/- 38 pixels in

the y coordinate (approximately 10% in each direction of the 1024 by 768 pixel display).” Given that eye movements were not constrained, how would variation of image position on screen serve to prevent low level adaptation? Note that ref 14 & 15 address this by changing body orientation (eye rotations are less likely than lateral eye movements), while Hummel et al 2012 (Perception) showed that low level adaptation of rectangles does not transfer to bodies. Also, recent study by Brookes et al (2018, Royal Society Open Science) showed that body adaptation generally has only a small low-level contribution. Also note that the variations of +/-51pix in 1024, or +/-38pix in 768 is approximately 5% in each direction (10% in total) for each dimension.

Authors’ response: We have used this approach as it is common practice in the face adaptation literature (Rhodes 2011), although we acknowledge that doing so may make little difference in terms of preventing low level adaptation. In terms of % variations, we meant 10% in the vertical and 10% in the horizontal direction, and have updated our text accordingly:

(p14) “approximately 10% in the vertical and 10% in the horizontal plane of the 1024 by 768 pixel display”.

10. “These are presented at baseline for the entire sample, and by randomization group.”
Please state where?

Authors’ response: Our apologies for the oversight in not including all variables at baseline. We have created a supplementary online table (2) containing those variables not in Table 1 to save space.

11. “The effects of training on judgement of own size were assessed using linear regression to compare post-intervention scores, adjusted for pre-intervention scores.”
How was this adjustment made? (Same question for other dependent variables where a similar adjustment is mentioned.)

Authors’ response: We have amended this as follows to clarify what we did:

P15: “The effects of training on judgement of own size were assessed using linear regression to compare post-intervention scores, adjusted for pre-intervention scores, by including the pre-intervention score as a covariate and the Stata command “regress”.”

12. “Randomisation group was treated as a categorical measure throughout.”
Repeated from 2 sentences earlier.

Authors’ response: We apologise for this oversight and have removed the first of these identical sentences.

13. “Analyses were repeated after adjusting for potentially confounding variables: number of images seen during training (as a measure of the amount of exposure participants had to the different images), time since last training [in tertiles] (in case any effects reduce over time), measured BMI (which influences perception of body size of others (18), and time spent using media

[in tertiles] (which may negate or reduce the effect of the study images).”

a. Why was the number of images seen during training not constant for all observers?

Authors’ response: Although all participants were asked to perform the task for 5 minutes twice a day, and were prompted with text reminders to do so, not all participants managed exactly this amount. Consequently, there was some variation in the number of images seen by each participant, which is why we included this term in the analysis. We have added the following to the Methods section to clarify this:

P15: “there was some variation since participants did not all comply with the instruction to complete the whole task, twice a day for five days”.

b. Why was time since last training/time spent using media divided into tertiles (rather than being treated as continuous variable)? How was this done, and what were the cut-offs?

Authors’ response: These measures were divided into tertiles (making 3 equally sized groups) because the measure was not normally distributed, and transforming the data in other ways did not normalise it.

c. What were the details of the media use questionnaire?

Authors’ response: We did not use a questionnaire as such: participants were asked the following questions: “How many hours per day do you spend on social media (Facebook, Instagram, etc)?”, “Approximately how many friends do you have on Facebook?”, “Approximately how many people/accounts do you follow on Twitter?”, “Approximately how many followers do you have on Twitter?”, “Approximately how many people/accounts do you follow on Instagram?”, “Approximately how many followers do you have on Instagram?”, “How many hours per day do you spend watching TV?”, “How many hours per day do you spend reading fashion/beauty/gossip magazines (online or print)?”

14. “Primary Outcome Measures: Own Size and Satisfaction with Own Size”

Insert “VAS” to distinguish this from avatar own size dependent variable.

Authors’ response: We have done so.

15. “Explicit Measures”

The placement of this heading excludes the two previous sections (VAS own body size and satisfaction, and perception of others’ body size), yet in the methods section, these are deemed explicit measures.

Authors’ response: Thank you for highlighting this, we have moved the heading “Explicit Measures” to above “Primary Outcome Measures”.

16. “There were differences between training groups in the BMI’s of their “own size” avatars (test of trend $p=0.007$) and “ideal size” avatars (test of trend $p=0.03$)”

Authors’ response: This quote seems not to have an associated comment, so we are not sure what the reviewer wanted us to revise with regard to this statement.

17. “The use of a range of outcome measures meant we were not reliant purely on explicit measures which could be affected by demand characteristics.”

This statement might carry some weight if all DVs – those susceptible to demand characteristics and those that are not – showed significant effects. Besides, it is unclear which DVs are immune to demand characteristics.

Authors’ response: We acknowledge this point and have removed this sentence.

18. Reference #29 This is an abstract which appears to describe the same study as ref #16. Avoid citing the gray literature.

Authors’ response: We apologise for this inadvertent citation of grey literature and have removed it.

19. There are several relevant papers that have not been cited, such as a recent review of body adaptation plus several empirical papers

Challinor et al., 2017 (International Journal of Medical Research),

Hummel, Grabhorn, & Mohr, 2012 (Perception);

Mohr, Rickmeyer, Hummel, Ernst, & Grabhorn, 2016 (Perception).

Authors’ response: Thank you. We have added these citations to our paper as described below.

Challinor (2017) as one of the authors suggesting adaptation as the underlying mechanism here:

P23: “The study methodology does not allow definitive specification of the mechanism of action of change in perception of the size of others’ bodies. However, many authors have argued that this is an “adaptation effect” (10-12, 15, 35).”

P4: “Three studies (N=14, 16 & 29) found exposure to thinner-than-actual photographs of self or others led to participants subsequently judging a thinner-than-actual photograph of self as more accurate; those shown fatter-than-actual photos subsequently thought fatter-than-actual photos of self were more accurate (16-18).”

P26: “Similarly, using interventions like that used here in a population of people with a current eating disorder may also shift their perception of what body size is “normal”; further research as to whether this would reduce their dissatisfaction with their own size would be informative. Some work suggests

that perceived own size in those with an eating disorder is more easily shifted towards overweight (Mohr 2016)".

Reviewer: 2

Comments to the Author(s)

The paper provides a useful contribution to the literature, and is written in a clear and comprehensive way which easily allows replication, and does a good job in attempting to understand its unexpected results.

Authors' response: Thank you for this very positive comment about our paper.

Some minor adjustments could be made prior to publication, listed below:

METHODS:

1. It appears that the paper utilised implicit measures alongside explicit measures to address bias. As such, it would be useful to include a statement regarding this in the Methods section to make clear how the experiment addressed potential sources of bias.

Authors' response: We have clarified this as follows:

"We included additional exploratory measures in order to help address possible problems with demand characteristics of the study, thus reducing a potential source of bias, and to test whether any changes in reported beliefs led to changes in behaviour, or changes on implicit measures."

2. Page 10, paragraph 3: Please change "half way" to "halfway".

Authors' response: We have done so.

RESULTS:

3. If there was any missing data in the questionnaires, it would be helpful to include a statement for how this was addressed.

Authors' response: We have added the following statement to the results section:

P17: "In terms of missing data, one person did not attend follow up, and their results were not included in the analysis. We had missing data for one participant for perception of other's body size, and two participants for own and ideal body size, due to in-session problems with the programmes for running these tasks. Analyses for these measures were conducted excluding those participants with missing data on these measures. We were unable to retrieve data from three laptops on the

time elapsed since the last time the training was completed, and adjusted analyses were therefore run excluding these subjects.”

4. Page 18, lines 32,33, 37, & 37: There appears to be missing 2 references.

Authors’ response: Thank you for highlighting this oversight, we have added the references into this section.

INTRODUCTION/DISCUSSION:

5. It would be useful to emphasise the clinical relevance of the experiment, either in the introduction, discussion, or both. If viewing overweight/underweight bodies has an impact on our own body image, how might that be applied in a clinical context? The authors do state that interventions drawing on this research might have potential use in preventing/treating body dissatisfaction, but do not give any information as to what these interventions might look like. Emphasising the clinical relevance of the research will be useful for the reader to understand why this research and its findings are of importance

Authors’ response: Thank you. We have added the following to the Introduction:

P3: “Body dissatisfaction is a potentially modifiable target for both prevention and treatment.”

We have added a longer section to the end of the Discussion as follows:

P26: “This suggests that at a population level, changing the size of bodies that people are regularly exposed to in the media would shift their perception of what constitutes a “normal” and “ideal” body size. It is not clear from this study that doing so would reliably effect satisfaction with own size. Similarly, using interventions like that used here in a population of people with a current eating disorder may also shift their perception of what body size is “normal”; further research as to whether this would reduce their dissatisfaction with their own size would be informative. Some work suggests that perceived own size in those with an eating disorder is more easily shifted towards overweight (37).”

Appendix B

Centre for Academic Mental Health

Population Health Science

Oakfield House, Oakfield Grove

University of Bristol

27.01.20

Dear Lianne,

Re: Manuscript ID RSOS-190704.R1 ("Does repeatedly viewing overweight versus underweight images change satisfaction with own body size?")

Many thanks for giving us the opportunity to respond to the new reviewer's comments on our paper, which we have done below. We very much hope that RSOS will feel that the paper is now ready for publication.

Reviewers' Comments to Author:

Reviewer: 3

The manuscript describes a study looking at the effects of repeated exposure to under, over and 'normal' weight bodies on perception of own and other body size, and also own body satisfaction. The authors employ an elegant study design, with participants randomised to one of three groups and then undergoing a 7 days worth of training with different sized body images. Measures of others body size, own body size and body satisfaction were made pre- and post-training in two lab visits

The hypotheses and primary outcome measures were pre-registered, which is a strength.

There were no differences between the groups in bodily satisfaction - the primary outcome measure. The tasks used to measure changes in others vs own body size differed in their methodology which complicates the findings pertaining to perceptual shifts.

For others body size participants were presented with 90 images of women's bodies and asked to make explicit rating of whether each was under, over or normal weight. Using this task they showed that following training with underweight images, participants rated more images as overweight (and fewer as underweight). This result was mirrored following training with overweight images. This is consistent with the authors hypotheses, that training with over or underweight images will shift perception of body size in the opposite direction.

For perception of own body size participants were instructed to adjust an avatar until it matched their own body size. Here they found that training with underweight images made participants adjust their avatar to appear smaller (and vice versa for training with overweight images). While seemingly inconsistent with their hypothesis, and the findings for the perception of others bodies, in the revised version of the manuscript that I am reviewing the authors discuss the likely possibility that adaptation has shifted the initial perception of the avatar (to appear larger in the case of underweight training) and so participants adjust the avatar smaller to compensate for this. In fact, considering the differences in measurement across own and other body perception, long-term visual

adaptation could plausibly account for both of the effects measured.

This is actually very interesting and adds to a broad literature suggesting that adaptation aftereffects can persist for many days and can build up outside of standard laboratory settings (Carbon & Ditye, *Frontiers in Psychology*, 2012).

It is, in my view, permissible to keep the hypotheses laid out in the introduction as they are (i.e. not accounting for the effects of adaption in the initial perception of the avatar) and to describe the interesting perceptual shift explanation for these results in the discussion, as the authors have done. My only suggestion to improve the manuscript is to carefully re-read and then re-structure the last part of the discussion in this regard, as the writing here could be clearer and these findings could be contextualised more in light of the broader literature on body size adaptation (i.e. perhaps explain what an adaptation aftereffect is and take a sentence or two to explain some of the common findings in relation to body size adaptation).

Authors' response: Thank you for this helpful review. We have rewritten these aspects of the discussion as follows:

"The study methodology does not allow definitive specification of the mechanism of action of change in perception of the size of others' bodies. However, many authors have argued that this change may be an "adaptation effect" (10-12, 15, 35). Adaptation occurs when a subject has a prolonged exposure to a stimulus. They then experience aftereffects of this exposure when they subsequently view other, similar, stimuli. A classic example of this is of focussing on an object moving in one direction; stationary stimuli viewed immediately afterwards appear to be moving in the opposite direction (36). Adaptation effects in relation to body size describe that repeated or extended exposure to images of smaller bodies leads to participants experiencing a perceptual aftereffect of viewing subsequent images as larger, and vice versa. Such perceptual aftereffects could plausibly be the underlying mechanism here."

"With relation to our use of the Avatar to demonstrate perception of own size, we believe that our hypothesis failed to take into account the effect of the training on participant perception of the starting size of the Avatar. That is, according to body size adaptation effects as described above, the aftereffect of exposure to smaller images would be that the Avatar starting image would appear larger. Even if the participant viewed their own body as being larger in comparison to the images they had seen, the Avatar image would appear larger too, and, in order for it to make the Avatar better represent their own size, they would have to adjust it to make it smaller. We had not anticipated that the strong adaptation aftereffect on the Avatar size could outweigh any change in perceived own size in this way."

with thanks and best wishes,

Dr Helen Bould (on behalf of coauthors)